# Inverse Optimization Latent Variable Models for Learning Costs Applied to Route Problems

**Alan A. Lahoud**[1]    **Erik Schaffernicht**[2]    **Johannes A. Stork**[1]

[1]Center of Applied Autonomous Sensor Systems (AASS), Örebro University
{alan.lahoud, johannesandreas.stork}@oru.se
[2] Technology Transfer Center Kitzingen,
Technical University of Applied Sciences Würzburg-Schweinfurt
erik.schaffernicht@thws.de

## Abstract

Learning representations for solutions of constrained optimization problems (COPs) with unknown cost functions is challenging, as models like (Variational) Autoencoders struggle to enforce constraints when decoding structured outputs. We propose an Inverse Optimization Latent Variable Model (IO-LVM) that learns a latent space of COP cost functions from observed solutions and reconstructs feasible outputs by solving a COP with a solver in the loop. Our approach leverages estimated gradients of a Fenchel-Young loss through a non-differentiable deterministic solver to shape the latent space. Unlike standard Inverse Optimization or Inverse Reinforcement Learning methods, which typically recover a single or context-specific cost function, IO-LVM captures a distribution over cost functions, enabling the identification of diverse solution behaviors arising from different agents or conditions not available during the training process. We validate our method on real-world datasets of ship and taxi routes, as well as paths in synthetic graphs, demonstrating its ability to reconstruct paths and cycles, predict their distributions, and yield interpretable latent representations. The code is available at https://github.com/AlanLahoud/IO-LVM

## 1 Introduction

When learning latent generative representations, it is often necessary for inferred samples to satisfy specific constraints, such as forming paths in a graph between designated start and target nodes, consistent with the feasible set of an associated constrained optimization problem (COP). A major challenge is learning such models when the feasible set of solutions is discrete, as the gradients of these solutions with respect to the model parameters are zero almost everywhere, and therefore non-informative [1]. In this paper, we pose this problem as learning a latent cost representation of the COP from observed solutions.

State-of-the-art approaches for recovering underlying cost functions of COPs from observed solutions, e.g., structured decisions performed by agents, primarily address the non-informative gradient problem by either smoothing solver operations [21], interpolating COP solutions [28], perturbing the COP cost [7], or relaxing the COP [39] to match statistics of the observed behavior (i.e., solutions). However, these methods assume a single underlying cost, making them unable to directly learn from data of multiple different agents with different underlying cost functions. To correctly learn from real world data containing behavior from different agents, they require supervision through agent labels.

In this paper, we introduce IO-LVM, a novel approach for learning latent representations of COP costs that can recover observed COP solutions, specifically for route problems in graphs. Our approach

39th Conference on Neural Information Processing Systems (NeurIPS 2025).

does not assume a single underlying COP cost, allowing it to learn effectively even when multiple agents or context are represented in the data without labels. Similar to a Variational Autoencoder (VAE) [18], we use amortized inference and map into a meaningful and interpretable low-dimensional latent cost space. In contrast of ordinary VAEs, we guarantee that samples fulfill requirements of the feasible set (e.g., connected paths) by using a black-box COP solver in the generative step. To address non-informative gradient challenge posed by discrete solutions, we employ a gradient estimation technique based on perturbing the input of the black-box solver and the Fenchel-Young loss [7, 8].

Applied to path and route data, we can use IO-LVM to, e.g., generate new paths by selecting a latent cost and providing source and target nodes to the black-box solver, which ensures validity of the path. This allows us to infer how different agents would navigate between new source and target nodes. As the IO-LVM learns costs instead of the shape / geometry of solutions, a low-dimensional latent representation is often sufficient, offering easy interpretation and analysis such as clustering similar COP costs, denoising observed paths by finding a small number of representative paths, and detecting outliers and deviating behavior.

We state our contributions as follows: **i)** We introduce IO-LVM, a method that combines variational approximation techniques with COP solver gradient estimation to learn latent representations for the underlying costs of COPs based on observed decisions; **ii)** IO-LVM naturally constructs a meaningful and interpretable latent space, allowing for the reconstruction of observed path distributions without making assumptions about the distribution of inferred paths. Notably, the ability to recover distinct (e.g., multimodal) representations for the underlying costs enables the modeling of different agents making decisions; **iii)** We demonstrate the versatility of IO-LVM using both synthetic and real-world datasets, highlighting its potential for path analysis tasks such as naturally separating underlying costs into meaningful groups, reconstructing or denoising observed paths, and predicting paths for unseen start and target nodes. Our aim is not only to provide quantitative results but also to offer insights through visualizations of routes and latent variables.

## 1.1 Related Work

Obtaining useful gradients through optimizers for end-to-end learning is challenging and previous works focussed on convex solvers [3, 2] and linear or quadratic programs [11, 36]. However, these methods are mainly limited to continuous COP formulations and are difficult to extend to combinatorial problems such as the graph-based problems in our work with the non-informative gradients. More related to our work are efforts to differentiate through dynamic programming algorithms, such as dynamic programming differentiability [22], or more specifically, a differentiable version of the Floyd-Warshall algorithm to learn from observed paths in graphs [21]. Different to our work, their approach struggles with scalability as graph size increases.

Learning cost parameters from observed solutions is also done by Inverse Optimization [4, 32, 33], Inverse Reinforcement Learning, and Inverse Path Planning [37, 21]. Here, cost parameters are either global [21], linear [23, 38, 39, 24] or non-linear [13, 37, 12], and often learned with end-to-end gradient estimation by exploiting insight in the underlying optimization process or in some cases assuming a black box optimizer [29, 7]. In the later case, the Fenchel-Young loss, also used in this paper, has been demonstrated as a suitable way to match inferred and observed paths within a smooth and convex space [7]. Different to us, these methods typically assume a single underlying cost function or condition the cost on a given context, which may not capture the diversity of agent behaviors present in real-world scenarios.

Autoencoders [17] and Variational Autoencoders (VAEs) [18] are often used to learn lower-dimensional representations of data and there have been attempts to simplify solving COPs by learning better representations of the feasible solutions [6]. However, we observe in our work that VAEs do not reliable generate feasible solutions in complex route problems like TSP and shortest path. In contrast to VAE, IO-LVM uses a COP solver in the decoding step and thereby guarantees that generated samples are feasible solutions.

## 2 Preliminaries

Below, we recap the Evidence Lower Bound (ELBO) for deep latent variable models and introduce Fenchel-Young losses which are both fundamental for our approach.

**Evidence Lower Bound (ELBO).** The objective in latent variable models is to identify the latent variables $\mathbf{z}$ that best explain the observed data $\mathbf{x}$. However, it is well-known that directly computing the posterior $P(\mathbf{z} \mid \mathbf{x})$ is generally intractable. To address this, a variational distribution $q_\phi(\mathbf{z} \mid \mathbf{x})$ is introduced and learned with a lower bound objective (ELBO) [18, 31]. The ELBO makes a trade-off between accurately reconstructing the input data (the expected log-likelihood) using a model $p_\theta(\mathbf{x} \mid \mathbf{z})$ and adhering to the prior distribution $P(\mathbf{z})$ for the latent variables, i.e.,

$$l(\theta, \phi) = -\mathbb{E}_{q_\phi(\mathbf{z}|\mathbf{x})}\left[\log p_\theta(\mathbf{x} \mid \mathbf{z})\right] + \beta D_{\mathrm{KL}}\left(q_\phi(\mathbf{z} \mid \mathbf{x}) \,\|\, P(\mathbf{z})\right), \tag{1}$$

where $D_{\mathrm{KL}}$ is Kullback-Leibler (KL) divergence and $\beta$ is a balance factor [16, 9]. In our approach detailed in Sec. 3, we also use a learned $q_\phi$ (encoder), but replace the usual reconstruction loss with a Fenchel-Young loss (see below).

**Fenchel-Young Losses** Fenchel-Young losses are a versatile class of loss functions derived from convex duality theory, specifically leveraging Fenchel conjugates [8, 7, 5]. These losses generalize several common loss functions used in structured prediction tasks, such as cross-entropy and hinge losses, by formulating the learning problem as a regularized prediction task [8]. Fenchel-Young losses are defined as $l_{\mathrm{FY}}(\mathbf{y}, \mathbf{x}) = \Omega^{FC}(\mathbf{y}) + \Omega(\mathbf{x}) - \langle \mathbf{y}, \mathbf{x} \rangle$, where $\Omega(\mathbf{x})$ is a chosen regularization function, $\Omega^{FC}(\mathbf{y})$ is its conjugate, and $\langle ., . \rangle$ represents the inner product between two vectors. In our context, $\mathbf{x}$ represents a decision vector (e.g., a structured output), and $\mathbf{y}$ a cost vector; the loss measures how suboptimal $\mathbf{x}$ is under the cost structure implied by $\mathbf{y}$, and is minimized when $\mathbf{x}$ is the optimal decision under $\mathbf{y}$ and regularizer $\Omega$.

## 3 Inverse Optimization Latent Variable Model

In this section, we introduce the notations and problem definition, present IO-LVM in a general COP setting (Sec. 3.1), and discuss assumptions for path and cycle applications (Sec. 3.2).

**Notation and Problem Definition.** Our dataset $\mathcal{D} = \{(\mathbf{x}_i, \mathbf{p}_i)\}_{i=1}^N$ consists of structured decision vectors $\mathbf{x}_i \in \mathcal{X}$ in a constrained space $\mathcal{X}$, e.g., connected paths performed by agents in a graph, and corresponding problem requirements $\mathbf{p}_i \in \mathcal{P}$, e.g., start and target nodes for the path. We denote by $\omega$ a black-box solver for the COP that takes cost vectors $\mathbf{y}_i$ and problem requirements $\mathbf{p}_i$ to output an optimal COP solution $\hat{\mathbf{x}}_i = \omega(\mathbf{y}_i, \mathbf{p}_i)$, where the COP solved by $\omega$ is defined as $\mathrm{argmin}_{\mathbf{x} \in \mathcal{X}(\mathbf{p}_i)} \langle \mathbf{y}_i, \mathbf{x} \rangle$ which, although formulated with linear costs, can represent a wide variety of COPs, specially combinatorial ones [20]. The main goal is to model a low-dimensional representation of COP costs $\mathbf{y}_i \in \mathcal{Y}$ that leads to the observed decision vectors from $\mathcal{D}$. Concretely, we aim to estimate the posterior distribution $P(\mathbf{z} \mid \mathbf{x})$, where $\mathbf{z} \in \mathcal{Z} \subseteq \mathbb{R}^k$ is a latent vector in a space of dimension $k$.

### 3.1 IO-LVM Description and Learning

We learn a latent representation $\mathcal{Z}$ with amortized inference [19] using a nonlinear mapping $q_\phi$ to map samples $\mathbf{x}_i$ to the latent space $\mathcal{Z}$, and then reconstruct them back to the constrained space $\mathcal{X}$, similar to a VAE. Different from VAE models, where reconstruction is done by a decoder network, our reconstruction is non-trivial due to the constraints on the COP solution space $\mathcal{X}$. E.g., $\mathcal{X}$ contains valid paths between specific nodes in a graph. To achieve this, we define our reconstruction as a composition of functions $g_\theta \colon \mathcal{Z} \to \mathcal{Y}$ and $\omega \colon \mathcal{Y} \times \mathcal{P} \to \mathcal{X}$, where the former is a nonlinear map parameterized by $\theta$, and the latter is a solver that is potentially non-differentiable. This sequence of transformations is visualized in Fig. 1.

To learn our IO-LVM, we adapt the VAE's ELBO objective by changing the reconstruction loss (first term) of Eq. (1). We introduce the solver $\omega$ and a suitable distance measure $d$ in $\mathcal{X}$ resulting in the term $\mathbb{E}_{q_\phi(\mathbf{z}|\mathbf{x})}\left[d(\mathbf{x}, \omega(\mathbf{y}^\theta, \mathbf{p}))\right]$, where $\mathbf{y}^\theta := g_\theta(\mathbf{z})$. In contrast to VAE models, the black box solver $\omega$ in our reconstruction generally prohibits end-to-end learning with a common loss such as the Mean Squared Error. For this reason, we use the Fenchel-Young loss for $d$ by inducing perturbations in the input space of the COP. Consequently, our loss function is defined as:

$$l(\theta, \phi) = \mathbb{E}_{q_\phi(\mathbf{z}|\mathbf{x})}\left[l_{\mathrm{FY}}(\mathbf{y}^\theta, \mathbf{x})\right] + \beta D_{\mathrm{KL}}\left(q_\phi(\mathbf{z}|\mathbf{x}) \,\|\, P(\mathbf{z})\right). \tag{2}$$

By choosing $\Omega$ (in $l_{\mathrm{FY}}$) to be the conjugate of $\mathbb{E}_\epsilon[\min_{\mathbf{x} \in \mathcal{X}} \langle \mathbf{y} + \epsilon, \mathbf{x} \rangle]$, we rewrite (brief derivation in Appendix A) $l_{\mathrm{FY}}$ as

$$l_{\mathrm{FY}}^\epsilon(\mathbf{y}, \mathbf{x}) = \langle \mathbf{y}, \mathbf{x} \rangle - \mathbb{E}_\epsilon[\min_{\mathbf{x} \in \mathcal{X}} \langle \mathbf{y} + \epsilon, \mathbf{x} \rangle]. \tag{3}$$

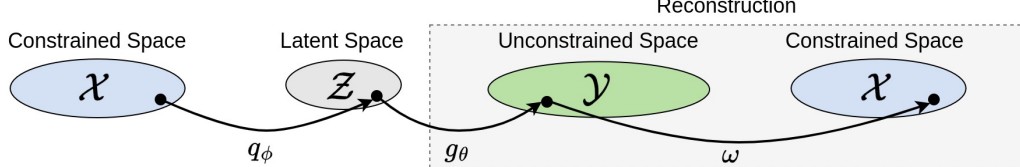

Figure 1: IO-LVM during learning: structured data is mapped from $\mathcal{X}$ to latent space $\mathcal{Z}$, then reconstructed in two steps: $\mathcal{Z}$ to unconstrained space $\mathcal{Y}$, and $\mathcal{Y}$ to constrained space $\mathcal{X}$ by a solver $\omega$.

The second term corresponds to the expected optimal cost under random perturbations of the cost vector $\mathbf{y}$. In Proposition 1 (see below), we justify this choice in the context of our problem setting by showing that, under mild conditions on the perturbation distribution $\mathcal{E}$, this formulation ensures that no solution is a priori favoured by the noise or problem structure, and solutions with the same cost impact gradients with the same probability. Specifically, we assume that the components of $\boldsymbol{\epsilon} \sim \mathcal{E}$ are independent, zero-mean, and identically distributed with finite variance, and that $\mathcal{E}$ is closed under convolution, e.g., Gaussian noise. In this case, the Fenchel-Young loss gradient with respect to $\mathbf{y}$-elements is analytically computed as $\nabla_{\mathbf{y}} l_{\mathrm{FY}}^{\boldsymbol{\epsilon}}(\mathbf{y}, \mathbf{x}) = \mathbf{x} - \hat{\mathbf{x}}_{\boldsymbol{\epsilon}}$, where $\hat{\mathbf{x}}_{\boldsymbol{\epsilon}} = \omega(\mathbf{y} + \boldsymbol{\epsilon})$, minimizing the Fenchel-Young loss if and only if $\hat{\mathbf{x}}_{\boldsymbol{\epsilon}} = \mathbf{x}$ [7]. This allows us to obtain gradient estimates w.r.t. the weights $\theta$, as the chain of gradients in the reconstruction block can now be written as

$$\nabla_\theta l_{\mathrm{FY}}^{\boldsymbol{\epsilon}}(\mathbf{y}^\theta, \mathbf{x}) = (\mathbf{x} - \hat{\mathbf{x}}_{\boldsymbol{\epsilon}}^\theta) \frac{\partial \mathbf{y}^\theta}{\partial \theta}, \tag{4}$$

where $\hat{\mathbf{x}}_{\boldsymbol{\epsilon}}^\theta = \omega(\mathbf{y}^\theta + \boldsymbol{\epsilon})$. Estimating the gradients in Eq. (4) is generally done in a Monte Carlo fashion, which is expensive due to the need of running the solver $\omega$ several times. To avoid this, we use the property of expectation linearity to rewrite the reconstruction loss in Eq. (3) as

$$\mathbb{E}_{q_\phi(\mathbf{z}|\mathbf{x})} \mathbb{E}_{\boldsymbol{\epsilon}} \left[ \left[ \langle \mathbf{y}^\theta, \mathbf{x} \rangle - \langle \mathbf{y}^\theta + \boldsymbol{\epsilon}, \omega(\mathbf{y}^\theta + \boldsymbol{\epsilon}, \mathbf{p}) \rangle \right] \right], \tag{5}$$

highlighting that the estimator is unbiased with a double expectation. This result allows us to use a stochastic gradient descent (SGD) method to learn $\theta$ and $\phi$, as described in Alg. 1. By leveraging SGD, the solver runs once (rather than several times in a Monte Carlo fashion) per data sample during training, reducing the computational cost per iteration.

**Algorithm details.** The algorithm details the steps in the training process using an encoder $h_\phi$ to model $q_\phi(\mathbf{z} \mid \mathbf{x})$, and a mapping $g_\theta$. Note that in step 1, the problem requirement sample can be leveraged into the encoder as additional information. Step 2 samples the latent value using the VAE re-parametrization trick [19]. In step 3, we include a transformation $\Phi$ to ensure that costs $\mathbf{y}^\theta$ fits to COP input space $\mathcal{Y}$. It is common that some COPs require their cost elements to be positive, for instance, which is generally fixed by ordinary activation functions $\Phi$. In step 4, we compute the COP solution given an inferred and

---

**Algorithm 1** One epoch of IO-LVM training process.

1: **Components:**
2:     - Encoder $h_\phi$; Decoder $g_\theta$.
3: **Input:** Dataset $\mathcal{D} = \{(\mathbf{x}_i, \mathbf{p}_i)\}_{i=1}^N$
4: **Output:** Trained model parameters
5: **for** each sample $(\mathbf{x}, \mathbf{p}) \in \mathcal{D}$ **do**
6:     **Step 1:** Encoding: $(\mu, \sigma) = h_\phi(\mathbf{x}, \mathbf{p})$.
7:     **Step 2:** Sample $\mathbf{z}$: $\mathbf{z} = \mu + \sigma \cdot \boldsymbol{\epsilon}, \boldsymbol{\epsilon} \sim \mathcal{N}$.
8:     **Step 3:** Map $\mathbf{z}$ to COP cost space: $\mathbf{y}^\theta = \Phi(g_\theta(\mathbf{z}))$.
9:     **Step 4:** Solve the COP using $\omega$, $\mathbf{p}$ and the inferred cost $\mathbf{y}^\theta$: $\hat{\mathbf{x}}_{\boldsymbol{\epsilon}}^\theta = \omega(\mathbf{y}^\theta + \boldsymbol{\epsilon}, \mathbf{p})$, where $\boldsymbol{\epsilon} \sim \mathcal{N}$.
10:     **Step 5:** Compute the loss as in Eq. (2).
11:     **Step 6:** Update the encoder and decoder parameters $(\phi, \theta)$ allowed by Eq. (4).
12: **end for**

---

perturbed cost. In step 6, the back-propagation is allowed due to the gradient estimator described in Eq. (4), bridging the gap of the non-differentiable COP solver.

## 3.2 IO-LVM Assumptions and Applications

We assume that the observed structured decision samples in $\mathcal{D}$ presented in Sec. 3 are optimal COP solutions, e.g., the taxi drivers solve a shortest path problem under their own underlying cost values. Therefore, the variations in the observed structured decision samples arises from differences in

valuations of COP costs. Below, we describe how we model the two problem domains used in experiments in Sec. 4.

**Path Planning.** For a fixed directed graph with edge set $E$, we can model a set of paths as a set of binary vectors $\mathcal{X} \subseteq \{0,1\}^{|E|}$ corresponding to edge usage. Here, $\mathcal{D}$ contains samples of paths ´in the graph. In this case, a common path requirement is $\{\mathbf{p} = (s,t) \mid s,t \in V, s \neq t\}$, defining start and target nodes of those paths. In this scenario, we assume there is an underlying set of edges costs for each data sample such that a Shortest Path Problem (SPP) solver (e.g., Dijkstra) $\omega$ recovers the observed paths.

**Hamiltonian Cycles.** For a fixed, either directed or undirected graph, with edge set $E$, we can model a set of cycles also as a set of binary vectors $\mathcal{X} \subseteq \{0,1\}^{|E|}$ corresponding to edge usage. Here, $\mathcal{D}$ contains samples of Hamiltonian cycles in the graph. We assume that there is an underlying set of edge costs for each data sample such that a Traveling Salesman Problem (TSP) solver recovers the observed cycle. Although path requirements could be added, we consider a null set in our experiments.

**Proposition 1.** *Suppose all feasible routes* $\mathbf{x} \in \mathcal{X}$ *have equal length, i.e., the same number of edges. Let* $\boldsymbol{\epsilon} \in \mathbb{R}^{|E|}$ *be a random perturbation vector with independent, zero-mean components, each with identical variance* $\sigma^2$*. Then, the perturbed cost vector* $\mathbf{y} + \boldsymbol{\epsilon}$ *induces a distribution over* $\mathcal{X}$ *in which all **solution costs** have equal expected value and equal variance, preventing bias toward specific solution in* $\mathcal{X}$ *after perturbation, and paths with the same cost impact gradients with the same probability.*

This condition holds exactly for Hamiltonian cycles, where all feasible solutions have the same length by definition, and serves as a reasonable approximation in our real-world path planning applications, where the graph is embedded in a space with relatively uniform node spacing and no significant shortcuts. Therefore, the corresponding Fenchel-Young loss chosen in 5, defined via the additive perturbation, is a good choice for both experimental settings. When this is not the case, alternative perturbation methods would be required. A proof for Proposition 1 is provided in Appendix B.

Note that in both applications, *as in general discrete problems*, the COP can be formulated with a linear objective [20]: $\hat{\mathbf{x}} \in \text{argmin}_{\mathbf{x} \in \mathcal{X}} \langle \mathbf{y}, \mathbf{x} \rangle$, fulfilling the requirement for the gradient estimator. Importantly, as in other works in structured prediction [7, 28, 21], IO-LVM is limited to problems for which fast solvers $\omega$ exist, since the solvers are the computational bottleneck of the training process.

### 3.3 IO-LVM Inference Tasks

Once the IO-LVM training process is complete, we can perform inference tasks in the above applications. IO-LVM allows reconstructing paths from parts of the low-dimensional latent space using again, as in the training step, a composition of the learned decoder and the solver, i.e., $\Phi(g_\theta)$ and $\omega(.)$, allowing us to observe different patterns reconstructed in the path space. By fitting a kernel density estimation (KDE) into the low-dimensional learned latent space, we can sample latent values and generate paths as mentioned above, leading to an inferred path distribution.

Selecting a proper $\beta$ during the training process is not only useful to mitigate the issue of posterior collapse or avoid overfitting as in $\beta$-VAEs [34], but in our case can also be useful to denoise a test path at inference time, i.e., remove uncommon patters in the observed test path by encoding it through the learned $h(\phi)$ and decoding it using $\Phi(g_\theta)$ and $\omega(.)$. In a similar direction, similarity metrics between a test path $\mathbf{x}$ and inferred paths from IO-LVM can also be used to detect if $\mathbf{x}$ is an outlier.

## 4 Experiments

The experiments focus on route problems in graphs using $\omega$ Dijkstra and a TSP solver for path planning and Hamiltonian cycles assumptions, respectively, as described in Sec. 3.2. Details of the IO-LVM training process are provided in Appendix F. We use four different datasets to evaluate IO-LVM in tasks like route reconstruction, path distribution prediction, facilitation to latent space analysis, and its potential for unsupervised learning tasks such as anomaly detection and denoising. The datasets used in the experiments are presented below, while further details are described in Appendix C.

**A) Synthetic Waxman Random Graph.** We generate a Waxman graph [35] with 700 nodes and 7230 edges; with three different cost functions for the edges costs $\mathbf{y}$, all of them on the basis of a

nonlinear function from unobserved features. We increased the costs of southern edges for agent 1 and of northern edges for agent 3, while agent 2 was unbiased in terms of south/north edges (in Fig. 2 and 5 it is possible to observe those preferences). We solved the SPP using Dijsktra for each agent cost multiple times on the basis of the generated edges costs plus a Gaussian noise, i.e., $\omega(\mathbf{y} + \boldsymbol{\epsilon})$, to generate multiple paths to $\mathcal{D}$. These paths are generated in two manners, one set with a single source-target pair (**Single S&T**), and another set with multiple pairs (**Mult. S&T**). In both cases, 6000 paths were generated, which 5000 were used for training.

**B) Ships Dataset.** Using Automatic Identification System (AIS) data from the Danish Maritime Authority [10], we use ship locations collected during three months. We project the locations to a grid graph with 2513 nodes and 8924 edges, resulting in a set of 2500 paths with different start and target nodes, where 2,000 were used for training.

**C) Taxi Dataset.** We use the Cabspotting dataset [27], which contains real-world taxi trajectories in the city of San Francisco, following a preprocessing approach to build a graph based on the available trajectories [21]. Unlike the original setup, we increase the number of nodes and edges to 1125 and 8022 to better reflect the realism of the actual trajectories, resulting in 101344 trajectories.

**D) TSPLIB.** From TSPLIB95 [30], we use *burma14* (14 nodes, 91 edges) and *bayg29* (29 nodes, 406 edges) graphs. Actual (underlying) edges costs $\mathbf{y}$ are generated using a nonlinear function incorporating unobserved features and Euclidean distances between nodes as offset. We create two versions for each graph, one using 3 unobserved features and another using 50 unobserved features. 3000 actual Hamiltonian Cycles (2400 for training) are generated with a TSP solver $\omega$ using the underlying costs $\mathbf{y}$ as input.

## 4.1 Qualitative Latent Space Analysis

In this section, we analyze the learned latent space $\mathcal{Z}$. The goal is to observe that paths generated by similar costs (e.g., coming from the same agent or similar context) are encoded close to each other in latent space after IO-LVM training. We also analyze how structure in the latent space influences reconstruction by sampling latent values and observing corresponding inferred routes.

Additional results of latent space analysis using the Taxi and TSPLIB datasets are reported in Appendix D.

**Synthetic Waxman Paths.** Fig. 2 shows the learned two-dimensional latent space for the **Mult S&T** dataset. The colors represent the agent that executed each path. Importantly, the agent identity was not provided during training. We observe that IO-LVM successfully disentangles the factors associated with the costs of the three different agents. For example, Agent 1 (red) follows several distinct paths with different start and target nodes, yet these paths are consistently mapped to a common region in the latent space, reflecting their shared underlying transition costs, that is, to avoid southern edges when possible. In contrast, the VAE latent space fails to exhibit this structure: paths performed by the red agent are spread in

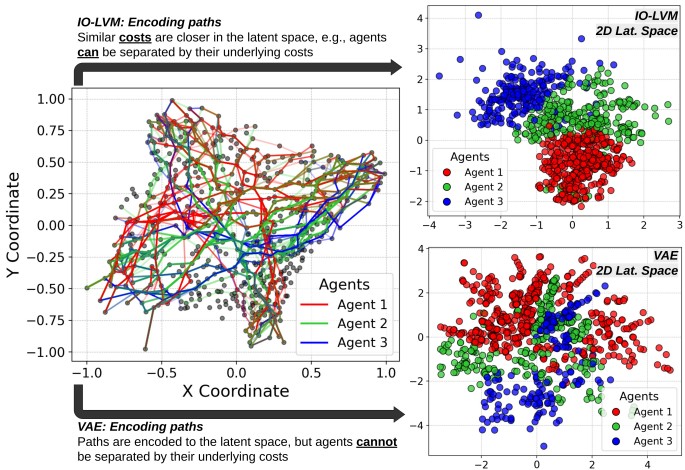

Figure 2: Left: Dataset (*Mult. S&T*) of paths from 3 agents (blue, red, green) with different underlying costs (+ noise). Right: Learned 2D latent spaces of IO-LVM (top) and VAE (bottom). IO-lVM groups data by cost while VAE groups data by path geometry.

different locations of the latent space. This suggests that VAEs are unable to disentangle the transition

cost structure inherent in the data. Illustrations of path reconstruction from different regions of the latent space are presented in Appendix D.

**Ship Paths.** The top-left chart in Fig. 3 illustrates the 2D latent space (3D latent space results are seen in Fig. 12 in the Appendix). Each hexagon in the right chart of Fig. 3 corresponds to a subspace of the latent space. For each hexagon, the average of the ships' width is plotted in color. Larger ships are less frequently found in the top-right corner of the latent space, leading to a low average ship width in that region. This is another example that IO-LVM was capable to capture unobserved factors within the latent space, i.e., the ship width information (provided by AIS) was not used during the training process.

We also analyze the reconstruction process through this dataset by sampling 20 latent values from Gaussians in the latent space and then computing the respective edges costs and paths. As seen in the right part of Fig. 3, neighbor latent values share a high number of edges in the graph (e.g., many path samples are the same). Moreover, an interesting pattern emerges: some regions of the latent space containing wider ships avoid the Oresund Strait when traveling from the east to the north part of Denmark even though it is the shortest path in terms of euclidean distance, as observed in the second column of the figure where ships prefer going through the Great Belt.

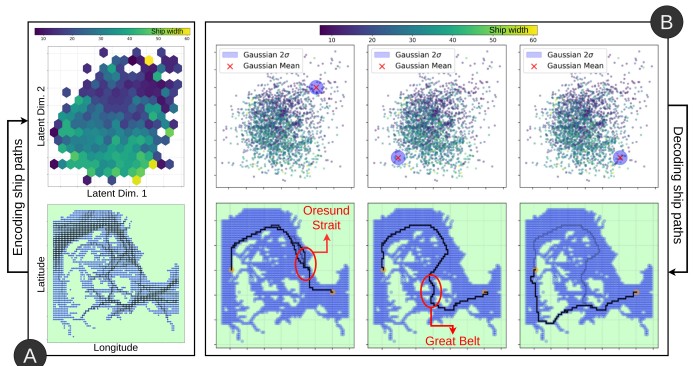

Figure 3: A) Ship paths (bottom-left chart in black) are encoded to the latent space using $q_\phi$ (top-left chart). Colors in the latent space represent the average ship width in each hexagon. B) Costs are sampled from Gaussians in the latent space (three top-right graphs), and respective paths are inferred given a hypothetical (non-existent in training paths) pair of start and target nodes (three bottom-right graphs).

## 4.2 Validating IO-LVM Reconstruction Quality

Table 1: Reported are the percentage of full reconstruction match (i.e., the reconstructed cycle matches perfectly with the target) calculated on the test set.

| Dataset / Methods | $\omega(\mathbf{y}_E)$ | VAE $(k = 2)$ | VAE $(k = 10)$ | IO-LVM $(k = 2)$ | IO-LVM $(k = 10)$ |
|---|---|---|---|---|---|
| burma14 (3 feats.) | 9.0% | $55.2 \pm 1.1\%$ | $78.4 \pm 0.6\%$ | $82.7 \pm 1.3\%$ | $\mathbf{91.0} \pm 0.7\%$ |
| bayg29 (3 feats.) | 4.5% | $15.4 \pm 1.3\%$ | $46.0 \pm 1.6\%$ | $37.3 \pm 7.3\%$ | $\mathbf{77.1} \pm 1.5\%$ |
| burma14 (50 feats.) | 1.3% | $9.8 \pm 1.3\%$ | $45.3 \pm 1.9\%$ | $29.6 \pm 2.6\%$ | $\mathbf{78.3} \pm 1.1\%$ |
| bayg29 (50 feats.) | 0.0% | $0.2 \pm 0.1\%$ | $3.4 \pm 0.5\%$ | $1.5 \pm 0.2\%$ | $\mathbf{31.9} \pm 3.5\%$ |

We evaluate reconstruction performance by comparing if the inferred output perfectly match the observed path (input to the encoder) with low latent dimensions on the TSPLIB datasets. Note that the task of reconstructing a perfect Hamiltonian cycle is challenging, since there are $13! \approx 6 * 10^9$ possible Hamiltonian cycles for the *burma14* dataset and $28! \approx 3 * 10^{29}$ for the *bayg29*. Results reported in Table 1 demonstrates that IO-LVM consistently outperforms VAE due to

Table 2: Reconstruction match calculated on the test set varying the training size for *bayg29* with 50 features.

| Methods / Train. size | 1000 | 10000 |
|---|---|---|
| VAE ($k = 10$) | 0.8% | 11.8% |
| VAE ($k = 100$) | 1.3% | 30.2% |
| IO-LVM ($k = 10$) | **20.8%** | **58.3%** |

its structured reconstruction process. As a naive reference, we include the $\omega(\mathbf{y}_E)$, a non-learning baseline where the TSP solution is computed using edge costs defined as the Euclidean distances between node positions. Although VAEs demonstrate to have good performance when the reconstruction is measured by *edge matching* instead of full reconstruction, as show in Table 4 (Appendix E), they do not guarantee feasible outputs with respect to the COP, leading to worse results in a full match comparison. Also, results in Table 2 shows that even by increasing the latent dimension of

VAEs, they are still far from the reconstruction power of IO-LVMs for the most challenging dataset (*bayg29* with 50 features). The gap is even higher when the number of training samples are smaller, as also shown in Table 2. Additional results on the variation of the number of latent dimensions are provided in Appendix E.

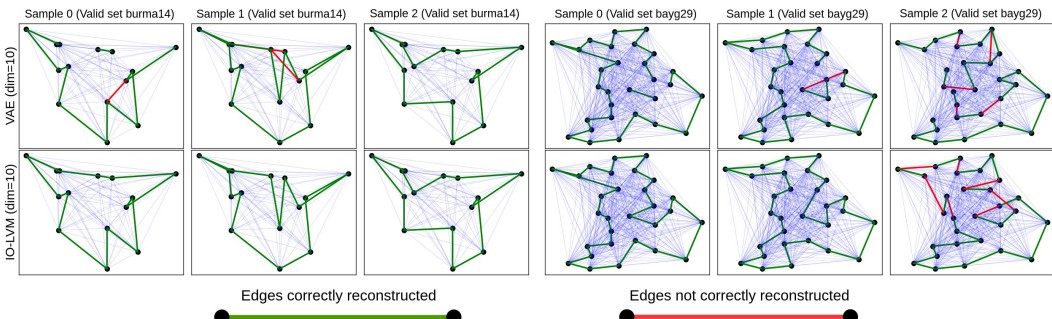

Figure 4: Each column illustrates an inferred sample for the Hamiltonian cycles experiment using VAE (top) and IO-LVM (bottom) on the first three samples of each dataset generated with 50 features. Green edges denote correct reconstructions relative to groundtruths, while red edges indicate false positives. VAEs might yield unstructured outputs. Groundtruths and other baseline results are available in the Appendix E, Figure 11.

Fig. 4 illustrates some reconstructed samples of IO-LVM against VAE , where the decoder outputs paths as binary edge usage indicators (i.e., probabilities converted to binary). In the figure, thick edges illustrate the reconstructed paths for the first three test samples of each dataset (*burma14* and *bayg29* created with 50 features). It can be seen from the figure that, different from VAEs, IO-LVM ensures that the output forms a valid Hamiltonian cycle due to the inclusion of a TSP solver in its processing loop. This happens even when the reconstruction is not fully correct (e.g., most right graph in in Fig. 4).

**Effect of varying $\beta$: Reconstruction versus Denoising** We analyze the effect of varying $\beta$ on two metrics in the *Single S&T* synthetic Waxman dataset: i) the number of distinct paths reconstructed by the decoder using the test dataset; ii) and the Intersection over Union (IoU) metric between observed and inferred edges usage during training. Figure 5 shows that as $\beta$ increases, the number of distinct paths decreases, indicating a denoising effect due to the diminished influence of the reconstruction loss. This results in the decoder reducing diversity of generated paths due to the posterior collapse. The IoU decreases with larger $\beta$, also reflecting a reduction in reconstruction power. In Figure 14 (Appendix E), we show a projection of noisy real-world taxi paths into similar but expected paths by reconstructing them using IO-LVM.

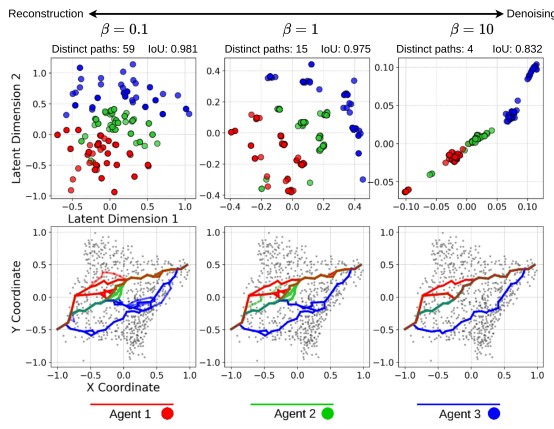

Figure 5: Lower $\beta$ yields more distinct paths reconstructed by IO-LVM, while a balanced or higher $\beta$ increases denoising.

## 4.3 Inference Results

**Path Distribution Prediction.** To measure the prediction quality of the overall test paths distribution, we used a kernel density estimator (KDE) in the learned latent space to estimate its probability density function and sample latent values from it. We evaluate the quality of predicted path distributions using two metrics: Jensen-Shannon divergence ($D_{\text{JS}}$, lower is better) and root mean squared error (RMSE, lower is better), both computed on edge usage frequencies between predicted and ground truth paths given fixed start and target nodes. For the Ship and Taxi datasets, these nodes are selected as the most frequent pairs in the data to ensure consistency in comparisons at inference time. Path

Table 3: Results on the prediction of paths distribution. Average and standard deviation of $D_{JS}$ and RMSE on edges usage over 20 runs are reported. NC denotes no convergence.

| Method | Single S&T | | Ship Data | | Taxi Data | |
|--------|------------|------|-----------|------|-----------|------|
| | $D_{JS}$ | RMSE | $D_{JS}$ | RMSE | $D_{JS}$ | RMSE |
| BO | $0.777 \pm 0.040$ | $3.678 \pm 0.299$ | $0.795 \pm 0.011$ | $6.44 \pm 0.07$ | $0.707 \pm 0.037$ | $5.70 \pm 0.09$ |
| PO | $0.058 \pm 0.008$ | $0.218 \pm 0.063$ | $0.164 \pm 0.007$ | $1.18 \pm 0.06$ | $0.439 \pm 0.003$ | $2.61 \pm 0.03$ |
| VAE | $0.060 \pm 0.005$ | $0.177 \pm 0.067$ | NC | NC | $0.599 \pm 0.020$ | $3.44 \pm 0.08$ |
| IO-LVM | $\mathbf{0.054} \pm 0.003$ | $\mathbf{0.151} \pm 0.034$ | $\mathbf{0.141} \pm 0.010$ | $\mathbf{0.96} \pm 0.11$ | $\mathbf{0.289} \pm 0.008$ | $\mathbf{1.56} \pm 0.10$ |

samples for the VAE and IO-LVM are obtained using KDE samples from their latent spaces, inferring their corresponding edges costs $\mathbf{y}^\theta$, and solving Dijkstra on them, i.e., $\omega(\mathbf{y}^\theta)$. We also compare the results against two other baselines: the Perturbed Optimizer (PO) baseline [7], which estimates gradients for structured prediction without modeling a latent space. We adapt PO to our setting by removing contextual inputs and reintroducing the training-time noise $\epsilon$ at inference to sample paths as $\omega(\mathbf{y}^\theta + \epsilon)$; and a naive BO baseline, where we minimize $RMSE(\mathbf{x}, \omega(\mathbf{y}_E + \epsilon^{BO}))$ through Bayesian Optimization where the distribution variance from which $\epsilon$ is sampled is a single decision variable. Results in Table 3 show that IO-LVM outperforms VAE, which fails to model constraints under limited data, e.g., in the Ship dataset, the sparse path count relative to the graph size prevented convergence (NC). IO-LVM also outperforms PO, whose naive sampling, i.e., $\omega(\mathbf{y}^\theta + \epsilon)$, fails to capture some characteristics of the true path distribution. The resulting inferred path distribution by IO-LVM of the Taxi dataset is illustrated with the blue paths of Figure 6 (top-left chart), and can be compared with the green observed paths in the test dataset with the same pairs of start and target nodes (top-mid chart).

**Outlier Detection.** We generate two path outliers in the taxi dataset as illustrated in the top-right chart of Fig. 6, one by using Dijsktra considering edge costs as Euclidean distances between nodes (purple path), and the other solving Dijkstra on learned costs but removing edges in latitude/longitude box, enforcing deviation (red path). To analyze if each of the observed paths (greens) and each of the outlier paths are actually outliers, we compute a low $\tau$-th quantile $Q^\tau$ (e.g., $\tau = 0.02$) of a distance measured between this test path $\mathbf{x}$ and the path distribution using inferred samples $\hat{\mathbf{x}}^\theta$ by IO-LVM, i.e., $Q^\tau_{z \sim \mathcal{Z}}[d(\mathbf{x}, \hat{\mathbf{x}}^\theta(\mathbf{z}))]$, a score reflecting how far each observed path lies from most of the paths in the inferred distribution. Although alternative distances $d$ could be chosen, we compute $d$ as the RMSE between the cost of those paths by sampling $N$ edges from the learned latent space, i.e., $d = RMSE([\langle \mathbf{x}, \mathbf{y}^\theta_i \rangle]^N_{i=1}, [\langle \hat{\mathbf{x}}^\theta, \mathbf{y}^\theta_i \rangle]^N_{i=1})$. This distance reflects how far two paths are from each other when projected on learned costs. A high quantile score $Q^\tau$ suggests that the path is unlikely under the inferred path distribution and therefore anomalous. We compute this score $Q^\tau$ for each path in the data

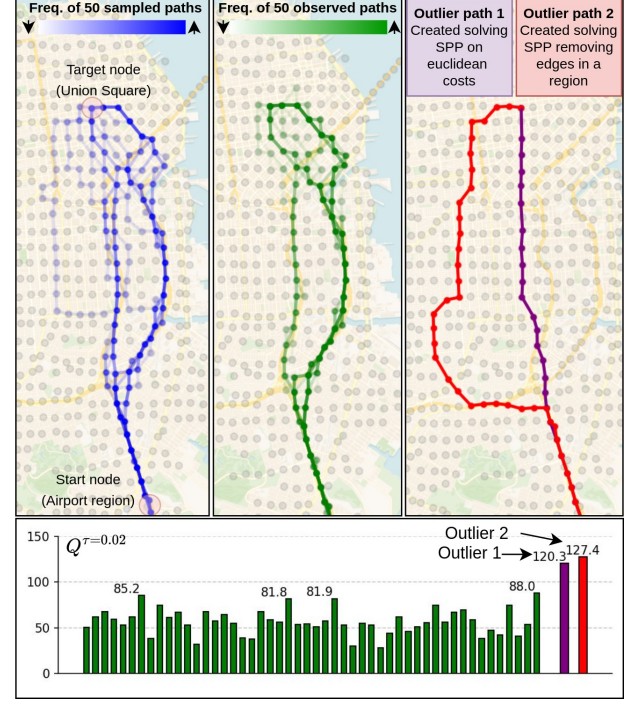

Figure 6: Taxi paths from a fixed start (airport region) and target (union square) nodes. Frequency of inferred paths by IO-LVM (blue), observed paths (green), and outlier paths (purple, red). Plots are generated with Open Street Map [25]. Quantile scores $Q^{\tau=0.02}_{z \sim \mathcal{Z}}[d(\mathbf{x}, \hat{\mathbf{x}}^\theta(\mathbf{z}))]$ for test paths $\mathbf{x}$ are reported for outlier detection.

(green) and the created outliers (purple and red) for $\tau = 0.02$. Results showing higher scores for outlier paths are reported in the bottom graphs of Fig. 6, and could also indicate the existence of other potential outliers in the test dataset (green bars).

## 5 Conclusion

This paper proposed IO-LVM, a novel approach for learning latent representations of COP costs, specifically for paths and cycles in graphs. The method leverages amortized inference and integrates black-box solvers within a probabilistic framework, allowing for the modeling of multiple agents and diverse behaviors in graphs. By employing a Fenchel-Young loss with perturbed inputs, it overcomes the gradient challenges in optimizing COPs, ensuring feasible and interpretable path reconstructions. The learned latent space captures meaningful structures, highlighting the model's characteristic to distinct agent behaviors, while maintaining accurate path reconstruction and prediction. Our method description is valid for a general set of COPs if gradient estimation is available.

## Acknowledgments and Disclosure of Funding

This work has been supported by the Industrial Graduate School Collaborative AI & Robotics funded by the Swedish Knowledge Foundation Dnr:20190128, and the Knut and Alice Wallenberg Foundation through Wallenberg AI, Autonomous Systems and Software Program (WASP).

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

# A  Perturbed Fenchel-Young brief derivation

For a regularized maximization problem, the Fenchel-Young loss is defined in [8] as

$$l_{\text{FY}}(\mathbf{y}, \mathbf{x}) = \Omega^{FC}(\mathbf{y}) + \Omega(\mathbf{x}) - \langle \mathbf{y}, \mathbf{x} \rangle$$

where $\Omega$ is a convex regularized and $\Omega^{FC}$ is its convex conjugate. By choosing the regularized $\Omega$ as the conjugate of the perturbed minimum cost, i.e., $\Omega^{FC} = \mathbb{E}_{\boldsymbol{\epsilon}}[\min_{x'}\langle \mathbf{y} + \boldsymbol{\epsilon}, \mathbf{x}' \rangle]$, the loss is rewritten as

$$l_{\text{FY}}(\mathbf{y}, \mathbf{x}) = \mathbb{E}_{\boldsymbol{\epsilon}}[\min_{x'}\langle \mathbf{y} + \boldsymbol{\epsilon}, \mathbf{x}' \rangle] + \Omega(\mathbf{x}) - \langle \mathbf{y}, \mathbf{x} \rangle$$

Since the only important terms for our gradient-based learning approach are the terms dependent on $\mathbf{y}$, i.e., we are only interested in gradients with respect to $\mathbf{y}$, and inverting the sign of the loss to be suitable to a minimization problem instead of a maximization problem, we simplify the loss for the gradient computation as:

$$l'_{\text{FY}}(\mathbf{y}, \mathbf{x}) = \langle \mathbf{y}, \mathbf{x} \rangle - \mathbb{E}_{\boldsymbol{\epsilon}}[\min_{x'}\langle \mathbf{y} + \boldsymbol{\epsilon}, \mathbf{x}' \rangle]$$

# B  Perturbed Fenchel-Young for Graph-based Applications

**Proof of Proposition 1.**  Let $\mathbf{y} \in \mathbb{R}^{|E|}$ be the edge cost vector, and let $\boldsymbol{\epsilon} \in \mathbb{R}^{|E|}$ be a random vector with independent, zero-mean components added to each edge. Let $\tilde{\mathbf{y}} = \mathbf{y} + \boldsymbol{\epsilon}$ denote the perturbed cost vector. For any path or cycle $\mathbf{x} \in \mathcal{X} \subseteq \{0,1\}^{|E|}$, represented as an indicator vector over edges, the perturbed cost is given by $\langle \tilde{\mathbf{y}}, \mathbf{x} \rangle$, representing the inner product between both variables.

We first show that the expected cost of a path remains unchanged under perturbation:

$$\mathbb{E}_{\boldsymbol{\epsilon}}[\langle \tilde{\mathbf{y}}, \mathbf{x} \rangle] = \mathbb{E}_{\boldsymbol{\epsilon}}[\langle \mathbf{y} + \boldsymbol{\epsilon}, \mathbf{x} \rangle] = \langle \mathbf{y}, \mathbf{x} \rangle + \mathbb{E}[\langle \boldsymbol{\epsilon}, \mathbf{x} \rangle] = \langle \mathbf{y}, \mathbf{x} \rangle,$$

since each component of $\boldsymbol{\epsilon}$ has zero mean and $\mathbf{x}$ is fixed.

Next, we compute the variance of the perturbed cost. The variance of the cost $\langle \tilde{\mathbf{y}}, \mathbf{x} \rangle = \langle \mathbf{y} + \boldsymbol{\epsilon}, \mathbf{x} \rangle$ arises from the perturbation:

$$\text{Var}[\langle \tilde{\mathbf{y}}, \mathbf{x} \rangle] = \text{Var}[\langle \boldsymbol{\epsilon}, \mathbf{x} \rangle].$$

Assuming the components $\epsilon_e$ are independent with identical variance $\sigma^2$, we obtain:

$$\text{Var}[\langle \boldsymbol{\epsilon}, \mathbf{x} \rangle] = \sum_{e=1}^{|E|} x_e^2 \, \text{Var}[\epsilon_e] = \sigma^2 \sum_{e=1}^{|E|} x_e = \sigma^2 \|\mathbf{x}\|_0,$$

where $\|\mathbf{x}\|_0$ denotes the number of edges in the path $\mathbf{x}$. Hence, if all $\mathbf{x} \in \mathcal{X}$ have equal length, the variance is the same for all paths.

As a direct consequence, the additive perturbation model induces equal expected cost and variance across paths or cycles when their lengths are equal. This condition holds exactly in our Hamiltonian cycle experiment, where all feasible cycles visit each node once and thus have fixed length. In the path planning experiment, although path lengths may vary, the uniform spatial layout in our real-world datasets (ship and taxi) and lack of large shortcuts make it reasonable to assume that most feasible paths between start and target nodes have similar lengths, while too lengthy paths are extremely unlike to be optimal even with perturbation. Consequently, the induced variance does not vary much, validating the use of this perturbation model in both settings.

# C  Datasets details

**Synthetic Waxman Random Graph**   We generate a Waxman graph [35] with 700 nodes ($\alpha = 0.05$, $\beta = 0.6$), where the probability of an edge between two nodes $u$ and $v$ is given by $P(u,v) = \alpha \cdot \exp\left(-\frac{d(u,v)}{\beta \cdot d_{\max}}\right)$, where we considered $d(u,v)$ as the Euclidean distance between nodes $u$ and $v$, and $d_{\max}$ is the maximum distance between of two nodes, consequently ending up in 7230 edges. We create three edge cost sets to simulate three different agents performing decisions to go from start and end nodes. The edge costs are based on Euclidean distances, with higher costs for the southern edges

for agent 1, and higher costs in the northern for agent 3, while agent 2 is not biased by the edges position. For each agent, we add a random noise in the cost elements $\mathbf{y}$ so the generated paths can be different from each other even within the same agent. The "observed" paths are generated by running the Dijkstra on the noisy edge costs. Two sets of 6,000 observed paths are generated: one with a single source and target pair (Fig. 2, top-left) and another with multiple source-target pairs (Fig. 2, bottom-left). In each of these sets, 5,000 paths are used for training IO-LVM and the baselines, while 1,000 are used for evaluation purposes. Further details on cost generation are provided in the code.

**Ships dataset**    We use the Automatic Identification System (AIS) data provided by the Danish Maritime Authority [10], considering latitude and longitude projected in a 2D space for simplicity. The analysis focuses on paths from the first week of the months January 2024, May 2024, and June 2024. Only paths that exceed a distance of 4 units (in latitude/longitude) in Euclidean space are included. A path is considered completed either when the ship speed approaches zero or when there is an abrupt change in its heading. In some cases, there are gaps in the latitude/longitude signals; when such jumps occur, we segment the data and treat them as separate paths. We created a grid graph with a distance of 0.09 units between adjacent nodes, focusing on the area where there are more route options to be taken, which in total led to 2513 nodes and 8924 edges. This resulted in approximately 2,500 ship paths, for which the first 2000 (in the order available from the data) is used for training. Note that this is the most sparse dataset, i.e. few paths in comparison with the graph size. This led, in our experiments, the VAE not being able to learn even after considerably more (e.g., 10x more) epochs than IO-LVM.

**Taxi dataset**    The data and the preprocessing follows exactly what is done in [21] with their available preprocessing code, except for the fact that we increase the density of the grid, so that the resulting in a bigger graph, consisted of 1125 nodes and 8022 edges and 101344 trajectories. We split the train and test datasets by randomizing the taxi drivers (anonymized driver id), where 70% was used for training and 30% for test.

**TSPLIB**    We use datasets from TSPLIB95 [30], a library of benchmark instances for the Traveling Salesman Problem (TSP) and related optimization problems. Specifically, we selected two graphs: *burma14*, which consists of 14 nodes representing locations in Myanmar, forming a complete graph with 91 edges, and *bayg29*, which consists of 29 nodes representing the coordinates of cities in Bavaria, Germany, forming a complete graph with 406 edges. To assign the actual edge costs $\mathbf{y}$, we uses the Euclidean distance between nodes as an offset, and design a nonlinear function that incorporates unobserved features to calculate edge costs. We generate two datasets for each graph, one considering 3 unobserved features (less complex), and one considering 50 unobserved features (more complex). The observed paths are generated as $\omega(\mathbf{y})$ without noise, where $\omega$ represents a TSP search solver. In all experiments with TSPLIB, even with different training size, we always used the same 600 test samples for a fair evaluation and proper comparisons. For the main results 2400 samples were used for training, while we had other experiments (see tables in the main paper) where we also used 1000 and 10000 training samples to understand the role of the training size in the reconstruction results.

# D  Additional Results: Latent Space Analysis

**Single S&T.** We sample 20 latent values from Gaussians in the latent space and compare distribution of paths generated by the composition of the decoder and solver. Reconstructed synthetic paths from the *Single S & T* dataset are shown in the bottom graphs of Fig. 7. It is observed that points closer in the latent space share a high number of edges in the graph. Additionally, as the variance increases, the number of distinct reconstructed paths grows, indicating consistency in the learned latent space. e.g., difference between third and fourth columns in Fig. 7.

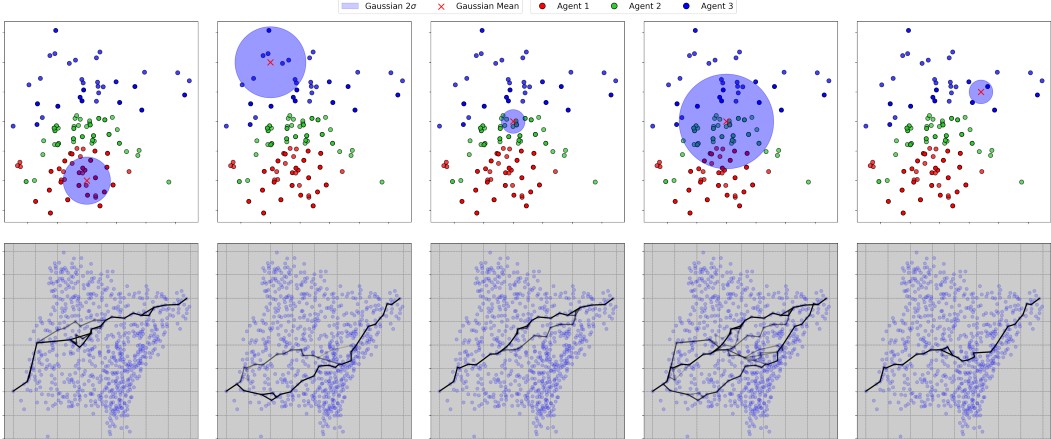

Figure 7: Reconstruction for the *Single S&T*. Top charts: region of samples from a Gaussian in the latent space. Bottom charts: corresponding generated trajectories. Blue agents has higher costs on edges in the north, while red edges has higher costs on edges in the south.

**Hamiltonian Cycles.** We select nine samples from the test set of the *burma14* graph, distributed across different regions of the latent space. They are organized into three groups of three samples each (see the top graph of Fig. 9). The corresponding paths that generated these latent values by the encoder are visualized in the bottom graph of Fig. 9. Paths with more edges intersection tend to be closer to each other in the latent space. We generalize this analysis by computing the Euclidean distance between all pairs of latent values versus the Manhattan distance based on edge usage (path choice) between those paths. For each group of sample pairs with a specific Manhattan distance, we calculate the average Euclidean distance in the latent space. The results, presented in Fig. 8, reveal that samples that are closer in the latent space are also closer in terms of edge intersection.

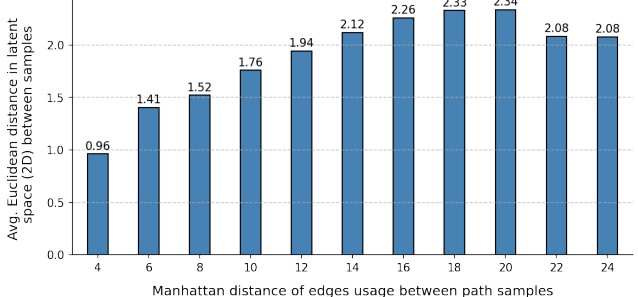

Figure 8: Relationship between Manhattan distance groups and the average Euclidean distance in the latent space. Samples with smaller Manhattan distances (i.e., paths with more similar edge usage) tend to have smaller Euclidean distances in the latent space.

# E  Additional Results: Inference

**Taxi Dataset.** We sample three different latent vectors from the learned latent space on the Taxi Dataset and compute three different set of learned edges costs, i.e., $\mathbf{y}^\theta = \Phi(g_\theta(\mathbf{z}))$. Then, we use Dijkstra as $\omega(\mathbf{y}^\theta)$ on those three set of edges to compute the shortest path given two set of nodes in the extremes of the graph on those learned edges. The resulting edges costs, normalized by the mean and standard deviation of a bigger sampling population of edges, and the resulting shortest paths, are illustrated in Figure 10.

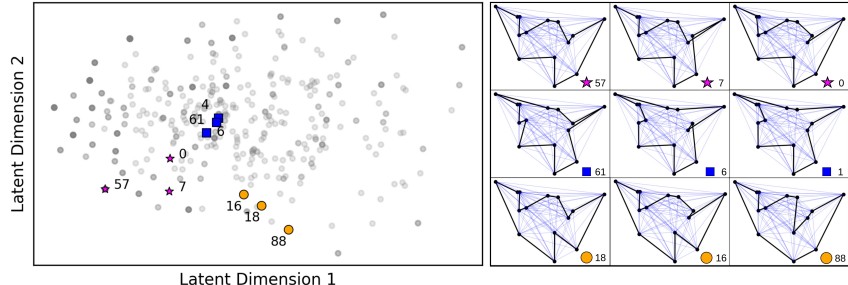

Figure 9: Visualization of the latent space and paths for the *burma14* graph. The top graph shows the latent space with nine manually selected samples. The bottom graphs display the corresponding paths for these samples.

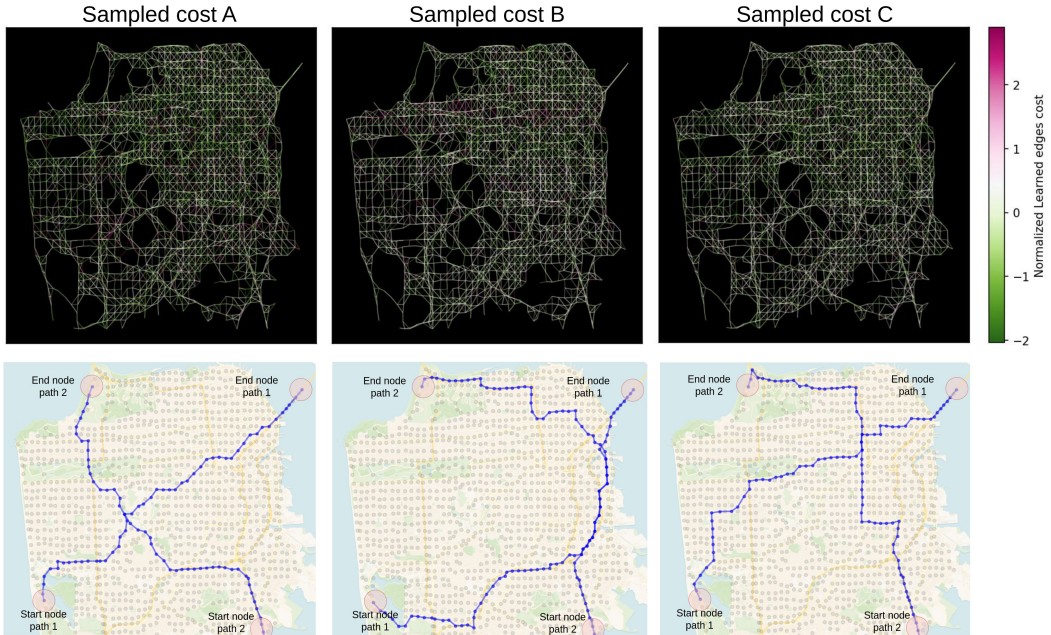

Figure 10: Three different set of edges costs from the learned latent space on the Taxi Dataset (top graphs, one per column). Corresponding shortest path given two set of nodes in the extremes of the graph on those learned edges (bottom graphs, one per column).

**TSPLIB reconstruction** The results in Table 4 are Recall of edges reconstruction in the TSPLIB datasets for both training and test path samples. IO-LVM is capable to reach almost 1.0 in all cases. VAEs can also do a good reconstruction when analyzing this metric. However, most of the mistakes in VAEs reconstruction leads to a non-structured output, i.e., a non Hamiltonian cycle, which can be observed with some reconstruction examples in Figure 11.

**Varying the number of latent dimensions**

In our experiments, we observed that for certain tasks, a very low number of latent dimensions was enough. For instance, in the Ship Dataset, attempting to add a third dimension to the latent space revealed that the second and third dimensions are highly correlated (Fig. 12), indicating that the third dimension is unnecessary. Conversely, in the Hamiltonian Cycles experiment, where 50 hidden features were used to generate edge costs with a complex relationship, increasing the latent dimensions proved beneficial in mitigating underfitting during the reconstruction process. This effect is illustrated in Fig. 13, which compares the performance of using 2 latent dimensions (left graphs) versus 10 latent dimensions (right graphs) for both the *burma14* (top graphs) and *bayg29* datasets. In the right-hand charts, we observe that using 10 latent dimensions achieves 100% Recall in the reconstructions for the training datasets and improves Recall for the test datasets compared to the

Table 4: Reported are the average Recall of edges reconstruction (edge matching) on the training (first row) and test (second row) sets for the Hamiltonian Cycles experiment.

| Methods | Lat. Dims | burma14 (3 dims) | bayg29 (3 dims) | burma14 (50 dims) | bayg29 (50 dims) |
|---------|-----------|------------------|-----------------|-------------------|------------------|
| VAE | 2 | $0.924 \pm 0.004$ | $0.805 \pm 0.007$ | $0.704 \pm 0.007$ | $0.572 \pm 0.026$ |
|  |  | $0.908 \pm 0.004$ | $0.798 \pm 0.003$ | $0.678 \pm 0.003$ | $0.555 \pm 0.028$ |
| VAE | 10 | $1.000 \pm 0.000$ | $0.995 \pm 0.002$ | $0.998 \pm 0.001$ | $0.916 \pm 0.025$ |
|  |  | $0.976 \pm 0.004$ | $0.957 \pm 0.003$ | $0.911 \pm 0.009$ | $0.799 \pm 0.033$ |
| IO-LVM | 1 | $0.900 \pm 0.003$ | $0.856 \pm 0.003$ | $0.731 \pm 0.004$ | $0.696 \pm 0.005$ |
|  |  | $0.890 \pm 0.007$ | $0.846 \pm 0.004$ | $0.724 \pm 0.005$ | $0.685 \pm 0.006$ |
| IO-LVM | 2 | $0.973 \pm 0.004$ | $0.908 \pm 0.016$ | $0.867 \pm 0.008$ | $0.786 \pm 0.010$ |
|  |  | $0.950 \pm 0.007$ | $0.883 \pm 0.013$ | $0.810 \pm 0.009$ | $0.723 \pm 0.005$ |
| IO-LVM | 10 | $\mathbf{1.000 \pm 0.000}$ | $\mathbf{0.999 \pm 0.000}$ | $\mathbf{0.999 \pm 0.000}$ | $\mathbf{0.997 \pm 0.001}$ |
|  |  | $\mathbf{0.985 \pm 0.002}$ | $\mathbf{0.971 \pm 0.005}$ | $\mathbf{0.960 \pm 0.004}$ | $\mathbf{0.900 \pm 0.004}$ |

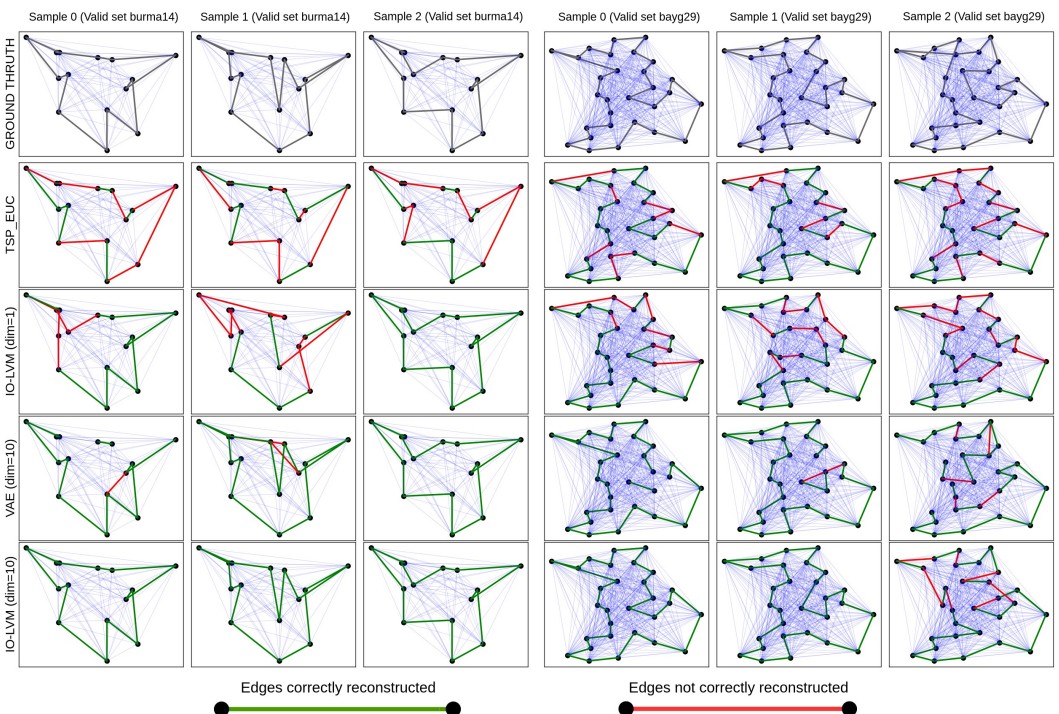

Figure 11: Each column illustrates an inferred sample for the Hamiltonian Cycles experiment. The first row represents the cycle observed in the test set. The second row represents a solution of the TSP using edges cost as euclidean distances (offset in the data generation process). Third and Fourth rows represent inference using VAE and IO-LVM with 10 latent dimensions. Green edges denote correct reconstructions relative to the groundtruth, while red edges indicate false positives.

2-dimensional case shown in the left-hand charts. However, with 10 latent dimensions, a slight overfitting emerges, which could be mitigated through more careful regularization and neural network architecture design. Addressing this was beyond the scope of our current study.

## E.1 Denoising Taxi paths

By encoding observed paths using the trained encoder of IO-LVM, and then decoding the corresponding latent value through the decoder + solver, it is possible to observe slight differences between the input and output. This is due to the well-balances $\beta$ chosen in the training process to avoid overfitting, leading to a denoising feature, i.e., removing uncommon patterns in an observed path. Figure 14

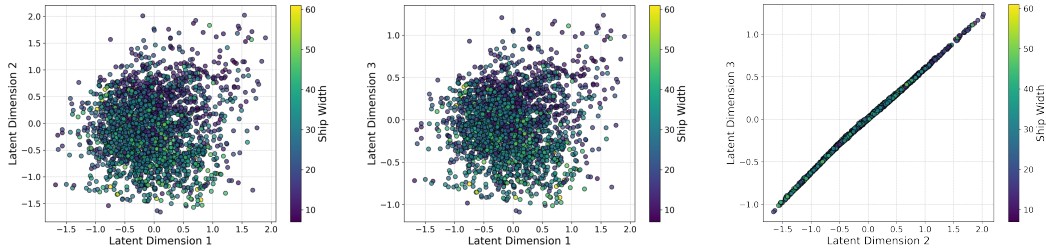

Figure 12: Latent space of ship trajectories using three dimensions. The right graph indicates that there is no need for a third latent dimension. Narrow ships are more concentrated in the top right corner of the two left graphs.

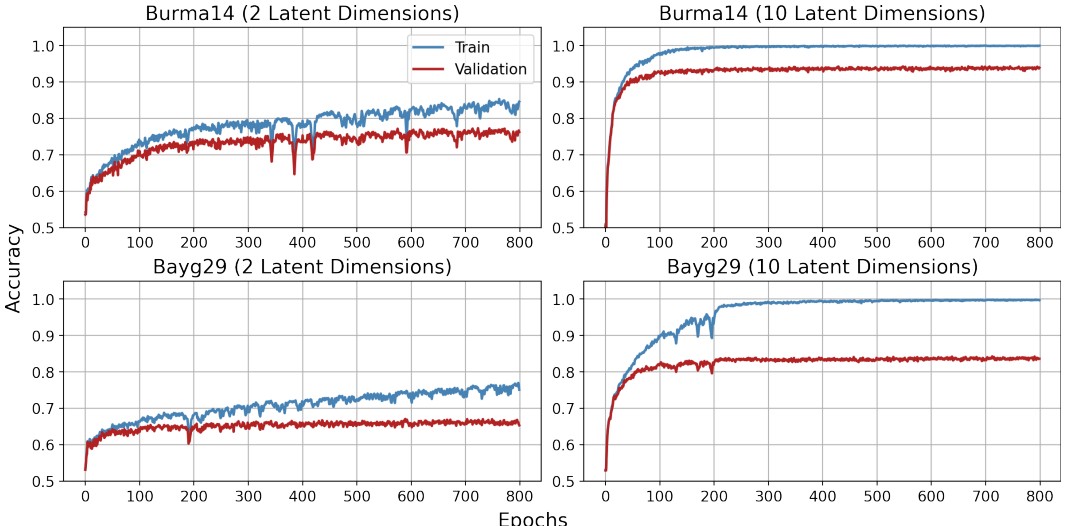

Figure 13: Comparison of training and validation curves for reconstruction performance using 2 latent dims (left) and 10 (right) for the *burma14* (top) and *bayg29* (bottom) datasets. Increasing the number of latent dims improves Recall for both training and validation datasets.

illustrates four samples of observed paths and its corresponding reconstruction. There are small differences in three of them, indicating less likely path decision patterns based on the training dataset.

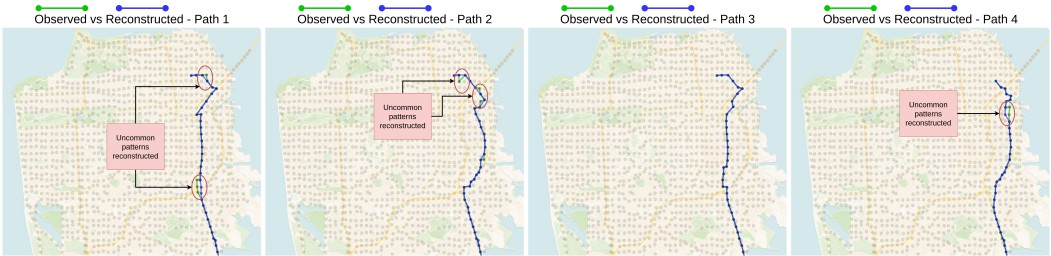

Figure 14: Four observed paths in the test dataset (green) are reconstructed (blue) by IO-LVM, eliminating potential uncommon patterns. When there is no uncommon pattern, the blue path overlaps the green path.

# F   Implementation details

IO-LVM is trained according to Algorithm 1. We used PyTorch [26] for the implementation. We consider a learning rate of 0.00004 and a batch size of 250 for the Waxman synthetic dataset, the Ship dataset, while the Taxi dataset and the Hamiltonian Cycle experiment uses a learning rate of 0.0001 and a batch size of 200. COPs are solved in parallel for the batches. The neural network architectures do not have any special implementations. The encoder architecture consists of a neural network with 4 hidden layers, each containing 1000 neurons, with ReLU activation functions in the hidden layers. The decoder architecture is the same as the encoder, but with a Softplus activation function to ensure that all edge costs remain positive. The RMSProp optimizer is used for the Synthetic and Ship datasets, while the AdamW optimizer is used for the Hamiltonian Cycles experiment. The experiments were run on a CPU due to the bottleneck introduced by COP solvers. The processor model used was the 13th Gen Intel(R) Core(TM) i7-13700KF, which has 24 cores. We use Dijkstra from networkx library in python [15] for solving SPP: and Ortools routing python library for TSP solutions [14]. For all the experiments, we run with a fixed and high number of epochs. The most time consuming experiment is with the Taxi dataset dueto the number of samples. We observed that after 100 epochs (around 2 days to complete with our resources) was enough for the model to converge. The other experiments can be finished in a maximum of few hours with no more than 500 epochs (the higher the better). However, for the VAEs baseline, more epochs are generally needed (we observed convergence after 1200 epochs for the synthetic experiment, for example). Further details are found in the provided code at `https://github.com/AlanLahoud/IO-LVM`

**Baselines.**   BO was performed to optimize a single variable i.e., the variance of the perturbation. 100 calls for each run was performed by varying the initial state. PO was trained by setting a pre-defined $\epsilon$ (Synthetic and Ship Datasets), as in related works, or by leveraging the mean of trained latent values in IO-LVM (Taxi Dataset), since the training process of PO is equivalent to IO-LVM with a single point in the latent space. Further details are found in the provided code.

