# OpenReview forum: "Inverse Optimization Latent Variable Models for Learning Costs Applied to Route Problems"
_NeurIPS.cc/2025/Conference — NeurIPS 2025 poster_

### Official Review · Reviewer_aXGz · 2025-06-23

**Clarity:** 4
**Significance:** 4
**Originality:** 4
**Rating:** 5
**Confidence:** 3

**Summary:**

Summary
This paper proposes a novel Inverse Optimization Latent Variable Model (IO-LVM) for learning implicit cost representations in constrained optimization problems (COPs) from observed optimal solutions. Unlike traditional Variational Autoencoders (VAEs), IO-LVM maps latent variables to cost function parameters and incorporates a black-box solver to ensure the generation of feasible, constraint-satisfying solutions. This design effectively addresses the challenges of non-differentiability and structural infeasibility common in structured output prediction. By leveraging Fenchel-Young losses and perturbed gradient estimators, the authors develop a principled training framework that enables unsupervised learning tasks such as path reconstruction, distribution prediction, interpretable latent space modeling, and anomaly detection.

The proposed approach is thoroughly evaluated on synthetic graphs, maritime shipping routes, taxi trajectories, and TSP benchmarks, covering multiple aspects such as reconstruction accuracy, distributional modeling, latent structure analysis, and inference performance. Comparative studies with representative baselines (VAE, Perturbed Optimizer, Bayesian Optimization) demonstrate the modeling capacity and practical potential of IO-LVM in unsupervised structured decision problems.

**Questions:**

The manuscript is clearly written and technically sound. The reviewer has no additional questions at this time.

**Ethical Concerns:**

["NO or VERY MINOR ethics concerns only"]

**Final Justification:**

After carefully considering the authors' rebuttal and the subsequent discussion, I find that the main concerns raised in the initial review have been adequately addressed. I am satisfied with their responses and have decided to raise my final rating.

**Limitations:**

This work primarily focuses on unsupervised cost inference in combinatorial optimization problems. Including a brief overview of recent advances in supervised CO solvers could further clarify the positioning and contribution of IO-LVM. Suggested references include (and potentially others) :

[1] DIFUSCO: Graph-based Diffusion Solvers for Combinatorial Optimization

[2] DISCO: Efficient Diffusion Solver for Large-Scale Combinatorial Optimization Problems

[3] T2T: From Distribution Learning in Training to Gradient Search in Testing for Combinatorial Optimization

**Paper Formatting Concerns:**

The reviewer has no concerns regarding the formatting of the paper.

**Quality:**

4

**Strengths And Weaknesses:**

Strengths

1. IO-LVM generates feasible solutions by mapping latent variables to cost vectors and invoking a black-box solver, which guarantees constraint satisfaction (e.g., valid paths or Hamiltonian cycles) and avoids the infeasible outputs often produced by VAE-based models.

2. The learned latent space naturally captures behavior patterns of different agents or preferences, enabling interpretable clustering and path preference analysis.

3. IO-LVM supports unsupervised inference tasks such as path denoising, anomaly detection, and distributional generation, filling a gap in current methods for unsupervised structured prediction.

Weaknesses

1. Despite adopting expectation linearization to approximate gradients, training still requires frequent calls to the black-box solver, which can be computationally intensive, especially on large-scale graphs.

2. The paper focuses exclusively on the unsupervised setting and does not discuss how supervised methods might perform when agent labels or ground-truth cost functions are available.


The writing, theoretical formulation, and empirical performance of this work are all strong. The reviewer is generally supportive of acceptance. However, a discussion on the relationship and comparative boundaries between supervised and unsupervised methods would further strengthen the paper. The reviewer would be willing to raise the rating upon incorporation of this clarification in the revised version.

---

> ### Author Rebuttal · Authors · 2025-07-27
>
> We thank the reviewer for the very positive assessment and for the constructive suggestions.
> We address the two substantive concerns below.
>
> ## 1. Training‑time solver calls
>
> **Reviewer’s concern.**
> > Frequent calls to the black‑box solver may be computationally intensive on large graphs.
>
> **Answer.**
>
> - **Why having a solver in the loop worth it?**
> First, the expectation‑linearisation technique reduces the cost to *one forward call per training sample*, rather than the multiple Monte‑Carlo calls used in classical perturbed‑optimizer methods.  Also, Feasibility of the yielded decision vector (e.g., paths, cycles) is guaranteed by construction.
> Pure generative models (VAEs, normalising flows, diffusion) must post‑process outputs with hand‑crafted heuristics (e.g., DIFUSCO details some heuristics in §3.5 of their paper) that are *problem‑specific* and do not generalize across COP families. In contrast, IO‑LVM inherits guarantees output feasibility as soon as a suitable COP solver is available for the formulated COP.
>
> - **Related work.**
> Frequent call of blackbox solvers are also presented in papers in well-known venues such as *Learning with Differentiable Perturbed Optimizers* (Berthet et al., 2020), *Differentiation of Black‑box Combinatorial Solvers* (Pogančić et al., 2020), and *DataSP: A Differential All-to-all Shortest Path Algorithm* (Lahoud et al., 2024). In SOTA, this is generally the price to pay to ensure outputs feasibility, unless the output structure (to ensure feasibility) is easier to model in other manners.
>
>
> ---
>
> ## 2. Supervised **vs.** unsupervised cost learning (and suggested references)
>
> **Reviewer’s concern.**
> > A discussion of boundaries between supervised and unsupervised methods is missing, and relate to references such as DIFUSCO, DISCO, T2T.
>
> **Answer**
>
> Supervised approaches such as DIFUSCO, DISCO, and T2T operate in a direct mapping setting. This means that these methods directly learn to map an instance to a *solution* (e.g., a path or cycle) without explicitly estimating COP costs. This has the advantage of not requiring solver calls during training, but the resulting model outputs may violate constraints unless additional decoding heuristics are applied (for example, DIFUSCO §3.5 proposes custom decoding for the specific problems they tackle). Another disadvantage of those methods is that they do not model a cost space, which is a key factor in our method for a simplified interpretability.
>
> IO‑LVM instead targets a mix of **supervised and unsupervised settings**. Supervised in a sense that **observed decisions** (e.g., paths or cycles) are available, and Unsupervised in the cost function space, where the underlying cost functions are latent. Our model explicitly learns a distribution over these latent cost functions and then uses a solver to produce solutions, guaranteeing feasibility by construction and allowing interpretable modeling of agent behaviours (and not only the output itself).
>
> If ground‑truth costs were available, as the reviewer pointed out, a solution could still be applied in a **two‑stage approach**:
> 1. learn to regress cost functions from data;
> 2. decode solutions using the solver.
> This would remove solver calls during training but potentially suffer from error propagation, since the cost predictor would not directly optimise decision quality (e.g., quality of inferred paths and cycles).
>
> We decided to add a summarized paragraph in the related work in the new version of our paper (to be updated in case of acceptance) comparing supervised learning methods for COPs (mostly combinatorial) in a direct mapping setting using some of the references suggested by the reviewers, as discussed above.
>
> ---
>
> We sincerely thank the reviewer for engaging in this constructive discussion. We hope that our responses regarding the boundaries of supervised vs unsupervised have helped to clarify the major concerns and we would be happy to provide any additional clarifications should further questions arise.

---

> > ### Comment · Reviewer_aXGz · 2025-08-01
> >
> > Thank you for your respond! My concerns have been addressed, and I will raise my score from 4 to 5.

---

### Official Review · Reviewer_4jFq · 2025-07-02

**Clarity:** 3
**Significance:** 3
**Originality:** 2
**Rating:** 5
**Confidence:** 3

**Summary:**

The paper proposes a latent variable model to learn representations of (unknown) cost functions from examples of the form (problem requirements, structured decision vector). The model, dubbed Inverse Optimization Latent Variable Model (IO-LVM), operates as follows:
* The pair is stochastically encoded to a latent variable, which is then decoded, similar to VAE
* Unlike a VAE, the decoded variable is not directly a sample. It represents costs for a constrained optimization problem (COP). The costs are passed to a black-box COP solver (with some perturbation to improve the efficiency of gradient estimation). The use of a solver in the reconstruction guarantees that the generated samples satisfy the problem constraints.
* A Fenchel-Young loss is used in place of the usual VAE reconstruction loss. This allows gradients to be estimated even though we cannot differentiate through the black-box solver.

The model is evaluated on two graph problems, routing and Hamiltonian cycles, over several datasets. Compared to traditional VAEs, IO-LVM is shown to produce better reconstructions and a more interpretable latent space. It is also demonstrated to be useful for anomaly detection.

**Questions:**

Do you have ideas for other problems to which IO-LVM could be applied? (No need to run additional experiments.)

**Ethical Concerns:**

["NO or VERY MINOR ethics concerns only"]

**Final Justification:**

Overall I think this is a solid paper that should be accepted. The authors have promised to add a brief derivation in their revision per my suggestion, and mentioned additional problems to which the method is applicable.

**Limitations:**

yes

**Quality:**

3

**Strengths And Weaknesses:**

Strengths
* The structure of the model guarantees that samples satisfy the constraints, which is a very useful property.
* Experiments demonstrate both the quantitative and qualitative superiority of IO-LVM to VAEs (and other baselines) for the tasks considered.
* The writing of the paper is generally clear and understandable.

Weaknesses
* The approach is limited to problems for which there is a suitable Fenchel-Young loss with analytic gradients. (Although, this covers a significant class of problems, e.g. those with linear objectives.) Experiments mostly only consider routing on graphs (also a bit about Hamiltonian cycles), which perhaps does not demonstrate the full utility of the model.
* To get equation (3) from the definition of $l_{FY}$ and the chosen $\Omega^{FC}$, one must argue that the $\Omega$ term disappears. The disappearing term is not mentioned at all, which was confusing. This is not a terribly long derivation, so I think it should be included in the appendix (or cite an existing derivation).

---

> ### Author Rebuttal · Authors · 2025-07-28
>
> We thank the reviewer for the very positive assessment and for the constructive suggestions. We discuss the raised points below.
>
> >> **Reviewer comment**: “The approach is limited to problems for which there is a suitable Fenchel-Young loss with analytic gradients (although this covers a significant class of problems, e.g., those with linear objectives). Experiments mostly only consider routing on graphs (also a bit about Hamiltonian cycles), which perhaps does not demonstrate the full utility of the model.”
>
> **Answer**
>
> We agree that our current experiments focus on routing tasks, and we chose these specific experiments for three main reasons:
>
> 1. **Visual interpretability**. Paths are easy to plot, making latent‑space diagnostics more understandable.
> 2. **Direct societal value**. Predicting routes helps to anticipate roadworks, identify urban mobility patterns, detect traffic anomalies, etc. The motivation of learning from demonstrations is more intuitive.
> 3. **Data availability**. Large‑scale, high‑quality demonstrations (GPS traces) are readily available, whereas labelled scheduling decisions, for example, are more difficult to find (to the best of our knowledge).
>
> However, the framework itself is not limited to routing. It can be applied, for example, to combinatorial problems with a linear cost structure, as correctly observed by the reviewer. This includes scheduling, resource allocation, and timetabling, for example. Note that even though the cost structure is not naturally linear, a problem reformulation could be used for linearization (more details in this matter is found in the discussion with reviewer eUDr). Those examples above could be potential problems to adress with IO-LVM if the aim is to "imitate scheduling" or "imitate resource allocation" and interpret those decisions in the COP cost space.
>
>
> >> **Reviewer comment**: Derivation of equation (3) from the definition of the fenchel-young loss.
>
> **Answer**
>
> We agree with the reviewer that this derivation will enhance the clarity of the paper, therefore, we will provide it in the revised version of the manuscript. Below are the steps that we will add to the Appendix:
>
> ---
>
> For a regularized maximization problem, the Fenchel-Young loss is defined in [8] as
>
> $\ell_{\mathrm{FY}}(y, x) = \Omega^{FC}(y) + \Omega(x) - \langle y, x \rangle$
>
> where $\Omega$ is a convex regularizer and $ \Omega^{FC} $ is its convex conjugate. We choose the regularizer $\Omega$ as the conjugate of the perturbed minimum cost:
>
> $\Omega^{FC} = E_{\epsilon} [ \min_{x'} \langle y + \epsilon, x' \rangle ]$
>
> so that
>
> $\ell_{\mathrm{FY}}(y, x) = E_{\epsilon} [ \min_{x'} \langle y + \epsilon, x' \rangle ] + \Omega(x) - \langle y, x \rangle$.
>
> Since the only important terms for our gradient-based learning approach are the terms dependent on $y$, i.e., we are only interested in gradients with respect to $y$, and inverting the sign of the loss to be suitable to a minimization problem instead of a maximization problem, we simplify the loss for the gradient computation as:
>
> $\ell_{\mathrm{FY}}'(y, x) = \langle y, x \rangle - E_{\epsilon} [ \min_{x'} \langle y + \epsilon, x' \rangle ].$
>
> ---
>
> We sincerely thank the reviewer once again for the opportunity to engage in this constructive discussion.

---

> > ### Comment · Reviewer_4jFq · 2025-08-08
> >
> > I would like to thank the authors for their explanations. My concerns are addressed.

---

### Official Review · Reviewer_eUDr · 2025-07-03

**Clarity:** 2
**Significance:** 2
**Originality:** 2
**Rating:** 4
**Confidence:** 4

**Summary:**

This paper presents a new method called IO-LVM for tackling inverse optimization problems, i.e., the problem of learning unknown cost functions from the solutions of constrained optimization problems. The work combines two techniques: latent-variable variational encoder models and gradient estimation via French-Young loss, to deal with two issues (1) learning of the underlying distributional representations of the cost vectors, and (2) non-informative gradient in discrete solution spaces, respectively. They present their framework’s versatility in route problems.

**Questions:**

1. It seems that the chosen FY loss is well-defined (as in your citation [7]) only when the COP is exactly a linear program with convex polytope constraint set. Otherwise, the linear operator (inner product) does not make sense, and the gradient may not be valid. I see no reason to define the FY loss this way if the objective of the COP for example is a general quadratic form of decisions $x^\top Y x$ where $Y$ is the cost matrix. I also doubt if the gradient derivation that works for the convex constraint set assumption, can also work here when the decision set is discrete. Can you confirm if this is the case? If yes, then I believe you need to specify it for a rigorous presentation, and you should explicitly discuss that your methods do not work for the general COPs.
2. I may not fully understand but the presented work does not perform or generalize inverse optimization; it effectively works on a related but not exactly same problem. I have no doubt It learns a latent representation of the cost function, and it is possible to sample a cost function out of the latent model. However, it is not clear if the framework can give the best prediction of the cost vector given a decision solution out of the COP, unless the black-box solver $\omega$ represents a invertible map that can recover $y^\theta$ from $\hat{x}_\epsilon^\theta$.
3. In Line 143, you mentioned that a benefit of your derived objective eq. (5) is that the solver runs once per data sample in SGD. I don’t see this as an advantage without any evidence. You need to compare with SGD that runs with solving multiple Monte-Carlo per data sample. It potentially yield better results as it offers a less noisy gradient that can lead to a steeper descent and require less iterations of training.

**Ethical Concerns:**

["NO or VERY MINOR ethics concerns only"]

**Final Justification:**

Most of my concerns are resolved during the rebuttal and discussions. I weakly recommend the acceptance if authors improve over  mathematical rigor in Section 3.

**Limitations:**

1. Following the first bullet in Questions, I believe the presented method is not sufficient for handling the general class of constrained optimization problems as claimed in Abstract and Intro.

**Quality:**

2

**Strengths And Weaknesses:**

**Strengths**
1. The paper tackles some well-motivated real issues in inverse optimization literature. The method IO-LVM presents a reasonable routine for solving these issues.
2. The method allows some interesting inference analysis by exploring the learned cost distribution, as a consequence of adapting the VAE models.

**Weaknesses**
The mathematical rigor needs some polishment in Section 3. This include undefined problem setups, undefined/inconsistent notations, etc.:

  - The symbol $\hat{x}\_{\epsilon}$ is never introduced. From Eq. (3) it looks like the gradient of the expectation term. Moreover, when you write $x=\hat{x}\_{\epsilon}(\theta)$,  $x$ are data while $\hat{x}\_{\epsilon}$ is a function of the trainable parameters $\theta$.  Thus, I believe it is more natural to flip the equation and write "if and only if we set $\theta$ such that $\hat{x}\_{\epsilon}(\theta) = x$".

  - In eq. (4), the LHS uses $y^\theta$ (with no $z$), while the RHS inconsistently uses $g\_{\theta}(z)$ and introduces notation $\hat{x}\_{\epsilon}^\theta$ with a new index $\theta$ for the first time.
  - A formal and mathematical formulation of the original optimization problem should be presented. No *mathematical language* asserts that you have the assumption that the dataset is heterogeneous with different agents solving a same optimization problem that has a common constraint space yet differs in personalized linear cost. I can raise more questions about the problem setting if not specified. For example, following Figure 2, do you assume that conditional on an agent, and a given problem requirement (fixed origin-destination), the cost is determined or random?
  - See first bullet in Questions.

These all impede a smooth reading and understanding of the methods.

2. Weak theory with insufficient and vague explanation.
  - Authors highlighted that the formulation ensures “equal variance and unbiasedness across the solution space” in Line 132, while in Proposition 1 the phrase used is “all solution costs have equal expected value and equal variance” What do these two phrases mean in mathematical language? The two expressions do not mean the same thing to me.
  - Furthermore, in Line 141, it is not clear what it means that eq. (5) highlights “the estimator is unbiased with a double expectation”. You probably mean eq. (5) is an unbiased estimator of the original objective in eq. (2).  Moreover, in Line 190, I cannot clearly see how the FY loss with additive perturbation yields a good choice.

---

> ### Author Rebuttal · Authors · 2025-07-28
>
> We sincerely thank the reviewer for the insightful questions and concerns raised.
> These comments/reviews were extremely valuable for improving the technical clarity and presentation of our work.
>
>
> **Regarding the symbol $\hat{x}_{\epsilon}$:**
> We have added an explicit introduction of $\hat{x}_{\epsilon}$ in Line 135 of the revised manuscript (to be updated in openreview in case of acceptance) as $\omega(y + \epsilon)$.  We also agree that reversing the equation order in the inline equation in Line 136 improves clarity, and we have applied this modification in the new version as well.
>
> **Regarding Eq. (4):**
> The reason we did not use $z$ on the LHS of Eq. (4) was to avoid inconsistency with the previous equations of $l_{FY}$ (e.g., Eq. (3)), while the use of $z$ on the RHS was meant to emphasize the encoding-decoding process.  However, we agree with the reviewer’s suggestion and, in the revised version, replace $g_{\theta}(z)$ by $y^{\theta}$, prioritizing clarity and self-content of eq.(4) itself over emphasizing the encoding-decoding pipeline.  We have also introduced $\hat{x}_{\epsilon}^{\theta}$ in Line 138, immediately after Eq. (4).
>
> **Regarding ambiguous wording in Lines 132 &Prop. 1:**
> The additive perturbation model, connected with Proposition 1, states that all solutions of equal length share the same expected cost and the same variance. This means that no path is a priori favoured by either the noise or the graph structure because solutions with the same expected cost is mapped to the same likelihood of being chosen with minimum cost (detailed in Appendix A).  Therefore, to enhance clarity, we have replaced the original sentence (lines 131/132):
> > “ensures equal variance and unbiasedness across the solution space”
> with:
> > “ensures that no solution is a priori favoured by the noise or problem structure.”
>
> Furthermore, we acknowledge that for other applications it can be more precise to replace the additive noise, as we state in the lines 191/192: “When this is not the case, alternative perturbation methods would be required”.
>
> ---
>
>
> ### Questions
>
> **1. Expressiveness of a linear cost model and validity of the Fenchel–Young (FY) gradient in discrete problems**
>
> We thank the reviewer for raising these point, as clarifying both aspects are important to support our claims.
>
> **(a) Expressiveness of a linear cost:**
> A linear cost with general non‑convex constraints is highly expressive, as combinatorial problems can be represented in this form.
> According to Korte & Vygen (*Combinatorial Optimization*),
> > “Virtually all combinatorial optimization problems can be formulated as integer programs …”
>
> which refers to integer linear programs in the book’s chapter (as we cite it in Lines 193–194 of our paper).
> In the case of a quadratic cost (in a discrete space), as the reviewer pointed out, one would introduce auxiliary variables and reformulate the problem with a linear cost before applying IO‑LVM. Such reformulations are outside the scope of this work but are standard practice.
>
> In case of continuous feasible spaces (QPs, LPs), simpler ways are available to compute gradients, such as differentiating KKT conditions, for example. However, as we introduced in lines 20 and 38, and mention in line 193, the focus of IO-LVM is on problems with discrete feasible sets.
>
> **(b) Gradients for non‑convex/discrete problems:**
> Reference [7] ("Learning with Differentiable Perturbed Optimizers (Berthet et al., 2020)") discusses how Fenchel-Young gradients apply when $\mathcal{X}$ is discrete. In fact, those gradients are useful mainly because they fit to discrete problems. We did not detail the steps to avoid confusion to our contributions. However, we agree that it is important to clarify this point and below we summarize the reason why they fit to discrete problems.
>
> The gradient $\nabla_y L_{FY}$ is needed to perform backpropagation during training, which depends on the solution of the COP solver. Because the COP cost is linear (and the reason why linear costs are expressive enough is discussed in the previous item), this solution is an extreme point of the convex hull of the original discrete feasible set. But the convex hull itself does not need to be explicitly described in our paper (i.e., the focus of our method is not to derive the fenchel young gradients) because the solution is the same, i.e., $\hat{x}_\epsilon$ is identical whether solving over $\mathcal{X}$ or $\operatorname{conv}(\mathcal{X})$ (convex hull). In simple words, although FY is defined under a convex feasible set, the trick is to consider that this convex feasible set is the convex hull of the problem of interest, which is initially defined under a discrete feasible set.
>
> ---
>
> **2. Scope of “inverse optimization”**
> Our use of the term “inverse optimization” in our paper follows the literature in recovering cost functions from observed decisions (e.g, Inverse Reinforcement Learning). We agree with the reviewer that, in general, the solution mapping from COP costs may not be invertible. However, this limitation is inherent to any inverse optimization approach when multiple cost functions yield the same observed decision. We would like to argue that this is not a specific limitation of IO-LVM.
>
> **3. Computational efficiency claim (Lines 142–144)**
> Our statement was intended to highlight **computational efficiency** rather than improved learning performance. Specifically, COP solver calls dominate training wall‑clock time.
> We thank the reviewer for the concern, and to enhance clarity, we have revised the text to:
> > “By leveraging SGD, the solver runs once (rather than several times in a Monte Carlo fashion) per data sample during training, reducing the computational cost per iteration.”
>
> ---
>
> We sincerely thank the reviewer once again for the opportunity to engage in this constructive discussion. We hope that our responses have helped to clarify the major concerns and we would be happy to provide any additional clarifications should further questions arise.

---

> > ### Comment · Reviewer_eUDr · 2025-08-02
> >
> > Dear authors,
> >
> > Thank you for your responses.
> > I have a bit more followups before I update my rating.
> > 1. I understand "no solution is a priori favoured by the noise or problem structure" as a consequence of Prop. 1, and your goal is to justify from this argument the conclusion that the additive perturbation is a good choice.  But it is not immediately clear to me the logic in between this statement and your conclusion. In addition, can you also explain what is the empirical consequence of a "bad" perturbation choce, or the consequence of some solutions in $\mathcal{X}$ is favored?
> >
> > 2. Thank you for the detailed clarification regarding my question 1. As you mentioned, to apply IO-LVM for the general COPs, it is required to perform some necessary transformation, or construct the convex hulls. Thus, it does not mean that your method can directly be applied to an arbitrary general COP. I would suggest clarifying better that what problems can be directly solved by your methods, such as the route problems, and how general COPs  through certain reformulation can benefit from your method.
> >
> > 3. Could you please answer the 3rd Weakness I raise? which also concerns a more explicit definition of your scope of work.
> >
> > Overall, my outstanding concern centers around a more rigorous and logical clarification and interpretation of your technical contributions and their scope. I have no trouble with the method itself. Thanks!

---

> > > ### Author Response · Authors · 2025-08-02
> > >
> > > We would like to thank again the reviewer for the comments and suggestions. Below we go through each of the three raised items.
> > >
> > > ---
> > >
> > > **Regarding perturbation**
> > >
> > > The assumption we made to defend that the additive perturbation is a good choice is that candidate solutions have approximately the same length (same number of edges to reach the goal/accomplish the task), which is always true for the Hamiltonian cycle exp due to its definition, and a possible assumption for start/target paths exp. If the assumption doesn’t hold, even the additive noise could be a “bad” choice in theory, and we would need to rely on empirical results when using it. Below is an example:
> > >
> > > Let us assume a situation that is far from the assumption above. For that, let us take an example of 3 possible paths from node A to node B, where each path has the same total cost c (given a fixed learning state, i.e., fixed theta), but path 1 has 1 edge, path 2 has 2 edges, and path 3 has 100 edges.
> > > With the additive perturbation in the edges, we have the following random variables as path costs:
> > >
> > > path 1: $c + \epsilon$
> > >
> > > path 2: $c + 2*\epsilon$
> > >
> > > path 3: $c + 100*\epsilon$
> > >
> > > This leads to path 3 to be chosen as the path with minimum cost more likely than path 1 and path 2 just because of the additive noise and the graph structure. To give an intuition, when a path has a variance much bigger than the others, almost half of the time it will be sampled as the minimum path (although other half is chosen as the maximum, but it doesn’t matter for the gradient estimation).
> > >
> > > Why do we need those paths to be chosen as minimum with equal probability (so that we can say it is a “good” perturbation)? Because when estimating gradients in Eq. (4), we want the expectation of $\hat{x}_{\epsilon}^{\theta}$ to be distributed equally through all those paths, i.e., paths with the same cost should impact the gradients with the same probability.
> > >
> > > One of the goals of proposition 1 is to point out the limitation of the additive noise. As we state in lines 191-192 and 130-131, an alternative perturbation method might be more suitable for different problem-settings.
> > >
> > > For these specific route experiments, an alternative could be to use a multiplicative noise when perturbing the edges, i.e., c*(1 + \epsilon), as it would not introduce variance bias in path costs. However, for that, we would need to investigate if an alternative gradient estimation would be required, i.e., if the change of perturbation method would have an impact on the fy gradient derivation, which we did not consider in the scope of this paper.
> > >
> > > ---
> > >
> > > **Regarding the COP expressiveness**
> > >
> > > Thank you for your suggestion. In the next item of this discussion, when adding the mathematical formulation of the optimization problem, we believe the scope of COPs are more clearly defined (please, see next item of our answer).
> > >
> > > Furthermore, in line 194 of our paper, as the reviewer suggested, we will add the suggestion of linearization in case of an initially nonlinear cost in the COP formulation, with the references *"Comparisons and enhancement strategies for linearizing mixed 0-1 quadratic programs (Adams et al., 2004)"* and "*Solving mixed integer bilinear problems using MILP formulations (Gupte et al., 2013)*" as examples.
> > >
> > > ---
> > >
> > > **Regarding weakness 3**
> > >
> > > Thank you for reminding us about this question.
> > > We agree that we can add the mathematical formulation of the original optimization problem in the problem definition. Therefore, in line 110-111 (Section 3), we will adjust to the following in the new revised version:
> > >
> > > … *$\omega(y_i, p_i)$, where the COP solved by $\omega$ is defined as $argmin_{x \in \mathcal{X}(p_i)}<y_i, x>$ which, although formulated with linear costs, can represent a wide variety of COPs, specially combinatorial ones [Korte & Vygen]. The main goal is to model a low-dimensional representation of COP costs of various agents $y_i \in \mathcal{Y}$* ...
> > >
> > > We will also redefine $\mathcal{X}$ as dependent on the requirements $p_i$ in line 107, to be consistent with the optimization problem formulation above.
> > >
> > > We believe that, with this, we address the initial lack of a *"mathematical language asserting that we have the assumption that the dataset is heterogeneous with different agents solving a same optimization problem that has a common constraint space yet differs in personalized linear cost"*.
> > >
> > > ---
> > >
> > > We thank once more the reviewer for the constructive feedback, as it helped us to provide a more polished version of our paper.

---

> > > > ### Comment · Reviewer_eUDr · 2025-08-05
> > > >
> > > > Thank you to the authors for your detailed and thoughtful responses.
> > > >
> > > > Your clarification regarding the rationale behind the perturbation is helpful. I agree that it is important to explicitly state that "paths with the same cost should impact the gradients with the same probability." This principle adds clarity to the underlying motivation.
> > > > That said, I still strongly encourage the authors to revise the exposition around Proposition 1. Improving the clarity and highlighting the significance of this result would strengthen the manuscript considerably.
> > > >
> > > > Given the authors' responses and I trust that the promised revisions will be incorporated, I am raising my score.

---

### Official Review · Reviewer_V5dK · 2025-07-03

**Clarity:** 3
**Significance:** 2
**Originality:** 3
**Rating:** 4
**Confidence:** 4

**Summary:**

The paper proposes IO-LVM, a generative model that learns latent cost representations of constrained optimization problems (COPs) from observed structured decisions, such as paths in graphs. By integrating variational inference with a black-box solver and using Fenchel-Young losses with perturbation-based gradient estimation, IO-LVM can generate feasible solutions and capture a distribution over cost functions, enabling modeling of diverse agent behaviors without labels. The method is evaluated on synthetic and real-world datasets, showing strong performance in reconstruction, latent space structure, and downstream tasks.

**Questions:**

See weaknesses above as well as the following questions.

* How does the model perform under suboptimal or noisy data?  (can you at least provide a discussion for this?)
* Can it scale with approximate solvers or caching?
* Is the latent space interpretable under interpolation?
* Can the method generalize beyond routing tasks to other COPs like scheduling?

**Ethical Concerns:**

["NO or VERY MINOR ethics concerns only"]

**Final Justification:**

Given the consensus of all reviewers, I think this is a good paper that should be accepted.

**Limitations:**

yes

**Quality:**

3

**Strengths And Weaknesses:**

Strengths

The proposed approach ensures constraint satisfaction via solver-in-the-loop decoding and handles multimodal behaviors by learning cost distributions. Theoretical grounding is solid, and experiments show clear advantages over VAEs and other baselines across multiple tasks. The learned latent space is interpretable and supports applications like denoising and anomaly detection.

Weaknesses

The method assumes optimal demonstrations and requires efficient solvers, limiting applicability to complex or noisy real-world tasks. Solver calls during training could be a bottleneck. Connections to more recent inverse RL or graph generative methods are not clearly stated.

---

> ### Author Rebuttal · Authors · 2025-07-28
>
> We thank the reviewer for their thoughtful feedback and constructive suggestions. Below, we address the questions and suggestions point by point.
>
> ---
>
> >> **Reviewer comment:**  "The method assumes optimal demonstrations.”
>
> **Answer**: Our framework **does not require global optimality under a single, unknown cost function**. Instead, we assume agent‑specific optimality. This means that each observed trajectory $x_i$ is optimal **for its own latent cost vector** $y_i$ over the feasible set $\mathcal{X}$.
>
> **Why this assumption is realistic?**
> Suppose the task is to perform a path from node A to node B. If two agents perform two distinct paths, both are considered optimal with respect to their own underlying (latent) costs. There is no assumption of a global optimal path. There is no assumption of a static COP cost (e.g., constant transition costs in the graphs).
>
> ---
>
> >> **Reviewer comment:**  "The method requires efficient solvers... Solver calls during training could be a bottleneck. Connections to more recent inverse RL or graph generative methods are not clearly stated."
>
> **Answer**:
>
> **Why having a solver in the loop worth it?**
>
> Note that by calling efficient solvers, feasibility is guaranteed by construction. Pure generative models (VAEs, normalising flows, diffusion) must post‑process outputs with hand‑crafted heuristics that are problem‑specific and do not generalize across COP families. Some of these methods, suggested by reviewer aXGz, for example, are discussed in the answer to reviewer aXGz. To summarize it here, in general (except for related work, that is discussed below), methods that attemps to solve the same problem without using COP solvers have the disadvantage of not ensuring feasibility or needs an extra layer that is problem-specific and tailored to map the output to the feasible set. In contrast, IO‑LVM inherits guarantees output feasibility as soon as a suitable COP solver is available, making our method more flexible to a wider variety of problems.
>
> **Related work.**
>
> Frequent call of blackbox solvers are also presented in papers in well-known venues such as "Learning with Differentiable Perturbed Optimizers (Berthet et al., 2020)", "Differentiation of Black‑box Combinatorial Solvers (Pogančić et al., 2020)", and "DataSP: A Differential All-to-all Shortest Path Algorithm (Lahoud et al., 2024)". In SOTA, this is generally the price to pay to ensure outputs feasibility across a wide variety of problem structures, unless the output structure (to ensure feasibility) is easier to model in other manners.
>
> ---
>
>
>
> >> **Reviewer question:** "How does the model perform under suboptimal or noisy data? (can you at least provide a discussion for this?)"
>
> **Answer**:
>
> This is a very important question. Our model is explicitly designed to remain reliable when demonstrations are noisy.  The main assumption is that the noise in the decision space comes from the variation of the underlying COP cost, e.g., agents perform tasks with distinct COP costs, and therefore it reflects to variations in the decision space (noise in paths, "near-optimality")). This assumption is stated in our paper in lines 167-168 in Section 3.2.
>
> ---
>
> >> **Reviewer question:** “Can the method scale with approximate solvers or caching?”
>
> **Answer**:
>
> This is also a very important question. We believe that scaling via approximate solvers and caching is an appealing research direction, and we appreciate the opportunity to discuss it. Although the current submission uses an exact solver at training time, the architecture is solver‑agnostic; replacing the exact oracle with an approximate or cached one is straightforward. However, the implications either on the results or in the time of convergence might harm, and it might depend on the bound guarantees of the solver near-optimality.
>
> We note that recent work in the broader "predict‑and‑optimize" literature already explores similar directions. For example, the work “Smart predict-and-optimize for hard combinatorial optimization problems” (Jantaya et al., 2020) shows that approximate or learned solvers can accelerate end‑to‑end training with a COP solver in the loop. We believe that extending such techniques to the *learning‑from‑demonstrations* setting we study here would be a promising research direction.
>
> ---
>
> >> **Reviewer question:** Is the latent space interpretable under interpolation?
>
> **Answer**:
>
> Yes! In Figure 9 (Appendix C) for example, we can observe that selected points are not in the training set, they are picked between training samples. Closer points in the latent space do not have much difference in edges usage in the graph, while distant points leads to more distinct cycle choices. A more general result is observed in Figure 8, where the statistics of the latent distance is correlated to the statistics of path divergence. Figure 7 (Appendix C) is another evidence, where we sample latent vectors using Gaussians, and the paths sampled from the same Gaussian are relatively similar (if the Gaussian variance is not too high). Note that, however, because of the COP nature, that it is still possible that small changes in the latent space result in very different solutions (e.g., a slight variation on edges costs might make a taxi driver to take the shortcut instead of the highway).
>
> ---
>
> >> **Reviewer question:** "Can the method generalize beyond routing tasks to other COPs like scheduling?"
>
> **Answer**:
>
> Yes! The framework is geenric to any COP formulated with a linear cost. Note that, as we discussed with reviewer eUDr, linear costs are powerful to formulate combinatorial problems. Linear costs are note only able to formulated problems like the ones we tackle in the paper but also many other combinatorial problems such as scheduling COPs (e.g., "The Schedule-Sequencing Problem (Bowman 1959)", "An ILP Model for Machine Scheduling (Wagner 1959)").
>
> **Why we focused on routing in this paper?**
>
> * **Visual interpretability.**  Paths are easy to plot, making latent‑space diagnostics immediately understandable.
> * **Direct societal value.**  Predicting routes helps to anticipate roadworks, identify urban mobility patterns, detect traffic anomalies, etc.
> * **Public data abundance.**  Large‑scale, high‑quality demonstrations (GPS traces) are readily available, whereas labelled scheduling decisions are more difficult to find (to the best of our knowledge). Note that we specifically need demonstrations / solutions for the same COP by different agents or under different costs. Such datasets are not easy to find in the real world.
>
> ---
>
> We once more thank the reviewer for the chance of having a fruitful discussion.

---

> > ### Comment · Reviewer_V5dK · 2025-08-05
> >
> > I have read authors' responses and the comments of other reviewers. I think this is a good paper that should be accepted.

---

### Comment · Area_Chair_6rQG · 2025-08-06

Dear Reviewers,

As the deadline for the author-reviewer discussion is approaching, we kindly request that you review the rebuttal at your earliest convenience if you haven't already done so, and engage in active discussion with the authors.

Please note that simply clicking the acknowledgement button without substantive input is not sufficient, as this may result in a flag for insufficient review. Additionally, delaying your response until the last moment will also trigger a flag.

Thank you for your understanding and for dedicating your valuable time to the review process.

---

### Decision · Program_Chairs · 2025-09-17

**Decision:**

Accept (poster)

**Comment:**

The paper proposes IO-LVM, a generative model that learns latent cost representations of constrained optimization problems from observed structured decisions. The approach leverages estimated gradients of a Fenchel-Young loss, propagated through a non-differentiable deterministic solver, to shape the latent space. Experimental results demonstrate the superiority of IO-LVM over VAEs and other baselines across several tasks.

The paper is generally well written and understandable. While reviewers raised some common concerns regarding the theoretical formulation, derivation, and applicability to other problems, the authors have provided clarifications that addressed these points. All reviewers noted that their comments were satisfactorily resolved. The authors are encouraged to incorporate the rebuttal clarifications into the final version to further improve the paper’s quality, for instance, by refining the exposition around Proposition 1 as suggested by a reviewer.

Given the overall positive evaluations from all reviewers, I recommend acceptance of this work.